# Insufficiency of 40S ribosomal proteins, RPS26 and RPS25, negatively affects biosynthesis of polyglycine-containing proteins in fragile-X associated conditions

**Katarzyna Tutak[1], Izabela Broniarek[1], Andrzej Zielezinski[2], Daria Niewiadomska[1], Tomasz Skrzypczak[3], Anna Baud[1]\*, Krzysztof Sobczak[1]\***

[1]Department of Gene Expression, Institute of Molecular Biology and Biotechnology, Adam Mickiewicz University, Uniwersytetu Poznańskiego 6, Poznan, Poland; [2]Department of Computational Biology, Institute of Molecular Biology and Biotechnology, Adam Mickiewicz University Uniwersytetu Poznańskiego 6, Poznan, Poland; [3]Center of Advanced Technology, Adam Mickiewicz University, Uniwersytetu Poznańskiego 10, Poznan, Poland

**\*For correspondence:**
anna.baud@amu.edu.pl (AB);
ksobczak@amu.edu.pl (KS)

**Competing interest:** The authors declare that no competing interests exist.

## eLife Assessment

In this **valuable** study, Tutak and colleagues set out to identify factors that mediate Repeat Associated Non-AUG (RAN) translation of CGG repeats in the FMR1 mRNA which are implicated in toxic protein accumulation that underpins ensuing neurological pathologies. The authors provide **solid** evidence that RPS26 may be implicated in mediating the RAN translation of FMR1 mRNA. This article should be of broad interest to researchers in the variety of disciplines including post-transcriptional regulation of gene expression and neurobiology.

**Abstract** Expansion of CGG repeats (CGGexp) in the 5' untranslated region (5'UTR) of the *FMR1* gene underlies the fragile X premutation-associated conditions including tremor/ataxia syndrome, a late-onset neurodegenerative disease and fragile X-associated primary ovarian insufficiency. One common pathomechanism of these conditions is the repeat-associated non-AUG-initiated (RAN) translation of CGG repeats of mutant *FMR1* mRNA, resulting in production of FMRpolyG, a toxic protein containing long polyglycine tract. To identify novel modifiers of RAN translation we used an RNA-tagging system and mass spectrometry-based screening. It revealed proteins enriched on CGGexp-containing *FMR1* RNA in cellulo, including a ribosomal protein RPS26, a component of the 40 S subunit. We demonstrated that depletion of RPS26 and its chaperone TSR2, modulates FMRpolyG production and its toxicity. We also found that the RPS26 insufficiency impacted translation of limited number of proteins, and 5'UTRs of mRNAs encoding these proteins were short and guanosine and cytosine-rich. Moreover, *the* silencing of another component of the 40 S subunit, the ribosomal protein RPS25, also induced repression of FMRpolyG biosynthesis. Results of this study suggest that the two 40 S ribosomal proteins and chaperone TSR2 play an important role in noncanonical CGGexp-related RAN translation.

## Introduction

Fragile X chromosome-associated syndromes are rare genetic diseases caused by dynamic mutations of the *fragile X messenger ribonucleoprotein 1* (*FMR1*) gene located on the X chromosome. The gene typically contains 25–30 CGG repeats in the 5' untranslated region (5'UTR). However, these triplet repeats are highly polymorphic and prone to expand, resulting in either a full mutation (FM; over 200 CGG repeats) or premutation (PM; 55–200 CGG repeats). On the molecular level, FM causes methylation of the *FMR1* promoter, leading to transcriptional silencing, loss of *FMR1* mRNA, and a lack of the main protein product, fragile X messenger ribonucleoprotein (FMRP), which is involved in modulating synaptic plasticity. An FM causes early onset neurodevelopmental fragile X syndrome (FXS), while PM is linked to many fragile X-associated conditions (FXPAC) including fragile X-associated tremor/ataxia syndrome (FXTAS), fragile X-associated primary ovarian insufficiency (FXPOI), and fragile X-associated neuropsychiatric disorders (FXAND). The estimated prevalence of PM is 1 in 150–300 females and 1 in 400–850 males. However, due to incomplete penetrance, approximately 1 in 5000–10,000 men in their fifties or later will develop FXTAS. In female PM carriers, random X-inactivation lowers the risk of FXTAS development (*Hagerman and Hagerman, 2016*; *Tassone et al., 2012*; *Jacquemont et al., 2004*). FXTAS is a late-onset neurodegenerative disease. Its pathology includes neuropathy, white matter loss, mild brain atrophy, and ubiquitin-positive inclusions in neurons and glia (*Hagerman and Hagerman, 2016*; *Greco et al., 2002*; *Greco et al., 2006*). Patients suffer from cognitive decline, dementia, parkinsonism, imbalance, gait ataxia, and tremors accompanied by psychological difficulties such as anxiety or depression (*Hagerman et al., 2018*; *Hagerman and Hagerman, 2016*). To date, no effective treatment targeting the cause, rather than the symptoms, has been proposed for any PM-linked disorders.

Three main molecular pathomechanisms are believed to contribute to FXTAS, FXPOI, and FXAND development (*Glineburg et al., 2018*; *Malik et al., 2021a*; *Hagerman et al., 2018*). First, high-content guanosine and cytosine nucleotides in the 5'UTR of *FMR1* cause co-transcriptional DNA:RNA hybrid formations (R-loops), which trigger the DNA damage response, thus compromising genomic stability, leading to cell death (*Loomis et al., 2014*; *Abu Diab et al., 2018*). Second, RNA gain-of-function toxicity induces nuclear foci formation by mRNA containing expanded CGG repeats (CGGexp) which form stable hairpin structures, and sequester proteins leading to their functional depletion (*Sellier et al., 2010*; *Sellier et al., 2013*; *Sellier et al., 2017*). Finally, mRNA containing CGGexp can act as a template for noncanonical protein synthesis called repeats-associated non-AUG initiated (RAN) translation. Protein production from expanded nucleotide repeats is initiated at different near-cognate start codons in diverse reading frames. The resultant toxic proteins contain repeated amino acid tracts, such as polyglycine (FMRpolyG), polyalanine (FMRpolyA), polyarginine (FMRpolyR), or hybrids of them produced as a result of frameshifting (*Todd et al., 2013*; *Wright et al., 2022*; *Glineburg et al., 2018*; *Kearse et al., 2016*). Notably, the open reading frame for FMRP starting from the AUG codon downstream to the repeats is canonically synthesized. In FXTAS and FXPOI, the most abundant RAN protein is the toxic FMRpolyG, which aggregates and forms characteristic intranuclear or perinuclear inclusions observed in patient cells and model systems (*Greco et al., 2002*; *Greco et al., 2006*; *Ariza et al., 2016*; *Todd et al., 2013*; *Sellier et al., 2017*; *Ma et al., 2019*; *Derbis et al., 2018*).

According to the current RAN translation model of RNA with CGGexp, the eIF4F complex and 43 S pre-initiation complex (PIC) bind to the 5'-cap of *FMR1* mRNA and scan through the 5'UTR until encountering a steric hindrance, a hairpin structure formed by CGGexp, or a nearby 5'UTR sequence. This blockage increases the dwell time of PIC and lowers initiation codon fidelity. As a result, the stalled 40 S ribosome initiates RAN translation at less favored, near-cognate start codons (ACG or GUG) upstream of repeats (in the FMRpolyG reading frame) or within CGG repeats (in the FMRpolyA reading frame; *Kearse et al., 2016*; *Green et al., 2016*). Unwinding stable RNA secondary structures appears to be crucial for initiating RAN translation, as several RNA helicases such as ATP-dependent RNA helicase DDX3X, ATP-dependent DNA/RNA helicase DHX36, and eukaryotic initiation factor 4 A (eIF4A) are involved in its regulation (*Linsalata et al., 2019*; *Kearse et al., 2016*; *Tseng et al., 2021*). Proteins indirectly involved in RAN translation can also contribute to RAN-mediated toxicity via pathways related to stress response and nuclear transport (*Green et al., 2017*; *Zu et al., 2020*; *Malik et al., 2021b*).

New insights into ribosome heterogeneity have explained rearrangements within ribosome components at different developmental stages or responses to environmental stimuli. This has led to

a better understanding of events that shape local ribosome homeostasis and affect the translatome (*Genuth and Barna, 2018*; *Shi and Barna, 2015*). For example, although ribosomes depleted of small ribosomal subunit protein eS26 (RPS26) stays functional in the cell, they are translating preferentially selected mRNAs (*Ferretti et al., 2017*; *Yang and Karbstein, 2022*; *Li et al., 2022*). Moreover, ribosomes containing small ribosomal subunit protein eS25 (RPS25) and Large ribosomal subunit protein uL1 RPL10A translate different pools of mRNA including those encoding key components of cell cycle process, metabolism, and development (*Shi et al., 2017*).

Recently, different proteins involved in RAN translation regulation reviewed in *Baud et al., 2022* were uncovered. However, mechanistic insight into this process remains unresolved. Given the toxicity of RAN translation products, identification of its regulating factors, which may serve as potential therapeutic targets to combat RAN proteins-related toxicity in fragile X-associated conditions, is essential.

In this study, we adapted an RNA-tagging technique to identify proteins natively bound to the RNA of the 5'UTR of *FMR1* with CGGexp that mimics mutated transcripts in PM carriers. Among tens of identified proteins, we focused on the RPS26 and investigated its involvement in CGGexp-related RAN translation. Previously, it was shown that RPS26 can interact with mRNA associated with PIC and regulate the translation of selected transcripts depending on their sequence, 5'UTR's length, and stress conditions (*Ferretti et al., 2017*; *Havkin-Solomon et al., 2023*; *Li et al., 2022*; *Yang and Karbstein, 2022*; *Pisarev et al., 2008*). By regulating RPS26 at the cellular level and its incorporation into the assembling 40 S subunit mediated by escortin TSR2 (*Schütz et al., 2014*; *Yang and Karbstein, 2022*), we found that insufficiency of this ribosomal protein has negative effect on the level and toxicity of FMRpolyG with no impact on encoding mRNA and FMRP. As FMRP is the primary protein product of the *FMR1* gene, it indicates that RPS26 depletion selectively modulates CGG-related RAN translation. Furthermore, by using a proteomic approach, we found that the number of proteins sensitive to RPS26 insufficiency was limited.

## Results
### Mass-spectrometry-based screening revealed numerous proteins interacting with the 5'UTR of FMR1 mRNA

To uncover new modifiers of CGGexp-related RNA toxicity, we used an RNA-tagging method coupled with mass spectrometry (MS) that allows for the in cellulo capture and identification of native RNA-protein complexes. The RNA bait, *FMR1 RNA* (used to pull down interacting proteins) consisted of the entire sequence of the 5'UTR of *FMR1*, which contained expanded, 99-times repeated, CGG tracts tagged with three times repeated MS2 RNA stem-loop aptamers (*Figure 1A*, *Figure 1—figure supplement 1*). These aptamers did not affect the RAN translation process as the synthesis of FMRpolyG was initiated from the near-cognate ACG start codon located upstream to CGGexp and terminated upstream to the MS2 aptamers (*Figure 1—figure supplement 1*). Given that the sequence of the 5'UTR of *FMR1* was guanosine and cytosine (GC)-rich (ca. 90% of GC content), we used an RNA sequence of the same length with GC content higher than 70%, *GC-rich RNA* (*Figure 1A*, *Figure 1—figure supplement 1*), as a control. Importantly, both of these sequences contained open reading frames similar in length but with differing start codons (ACG in *FMR1 RNA* and AUG in *GC-rich RNA*). Thus, they served as templates for protein synthesis (*Figure 1—figure supplement 1*). Together with RNA baits, the MS2 protein— showing high affinity towards RNA MS2 stem-loop aptamers—was co-expressed in HEK293T cells to immunoprecipitate natively formed RNA bait-protein complexes.

The MS-based screening identified over 500 proteins binding to two prepared RNA baits. Of those 172 proteins were identified in all technical replicates for FMR1 RNA bait (*Supplementary file 1, table 3*). The most enriched Gene ontology (GO) terms indicated that the majority of proteins binding to *FMR1 RNA* had RNA binding properties and were involved in ribosome biogenesis, translation, and regulating mRNA metabolic processes (*Figure 1A*, *Figure 1—figure supplement 1*, a complete list of GO terms is in the *Supplementary file 1, table 4*).

While searching for novel RAN translation modifiers, we focused on proteins enriched on *FMR1 RNA*. We applied a label-free quantification analysis, which allowed us to elucidate 32 significantly enriched interactors of *FMR1 RNA* compared to *GC-rich RNA* (*Figure 1B*, *Supplementary file 1*). Next, we preselected eight proteins based on their biological function and possible role in modulating translation, and using short interfering RNA (siRNA)-based silencing, we investigated the effect of

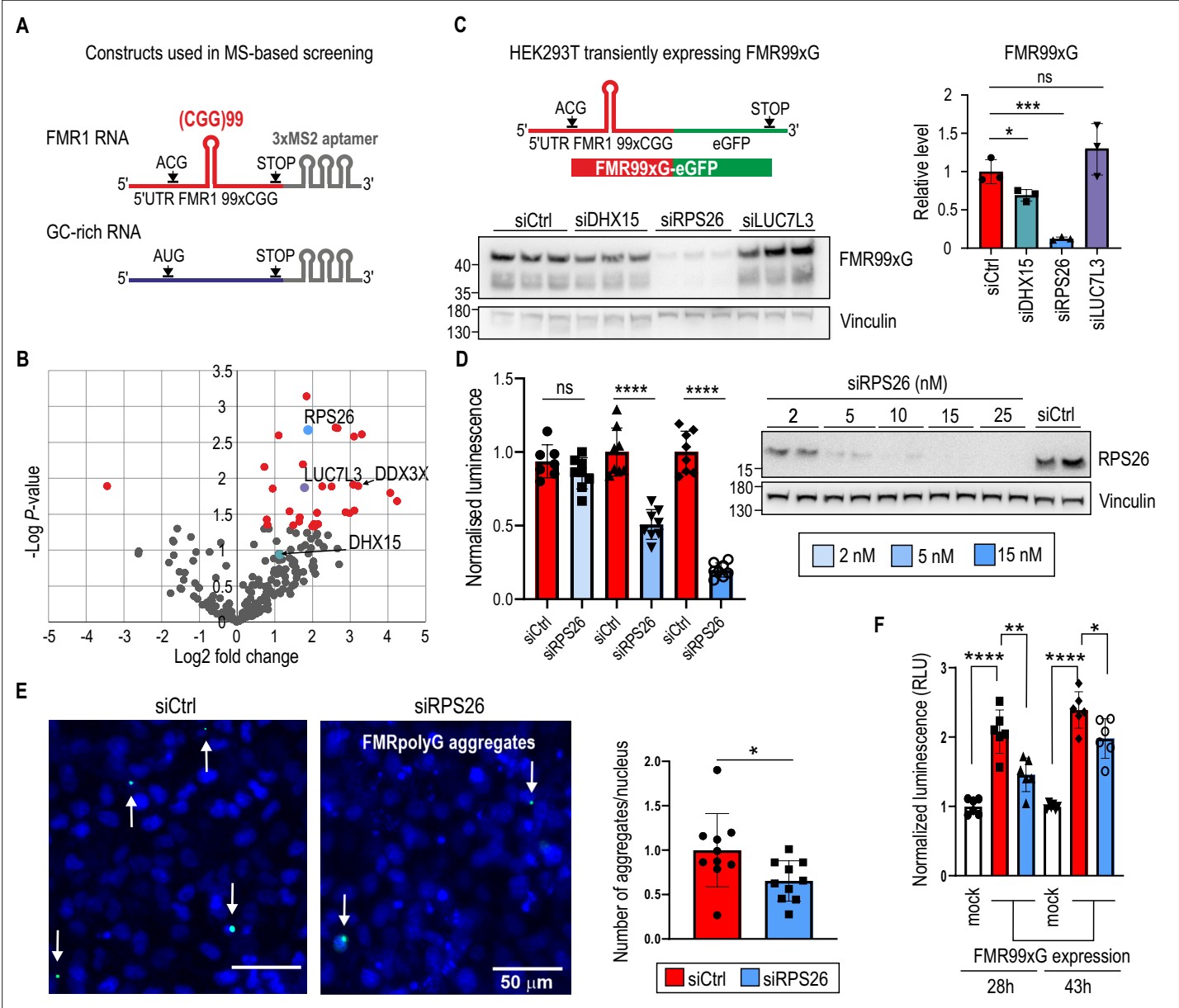

**Figure 1.** Mass-spectrometry (MS)-based screening revealed several proteins bound to *FMR1* RNA with expanded CGG repeats; some affect the yield of polyglycine-containing proteins, alleviating their toxicity. (**A**) Scheme of two RNA molecules used to perform MS-based screening. The *FMR1 RNA* contains the entire length of the 5' untranslated region (5'UTR) of *FMR1* with expanded CGG (**x99**) repeats forming a hairpin structure (red). The open-reading frame for the polyglycine-containing protein starts at a repeats-associated non-AUG initiated (RAN) translation-specific ACG codon. RAN translation can also be initiated from a GUG near-cognate start codon, which is not indicated in the scheme. The GC-rich RNA contains *TMEM107* mRNA enriched with G and C nucleotide residues (GC content >70%; similar to *FMR1* RNA) with the open-reading frame starting at the canonical AUG codon (blue). Both RNAs are tagged with three MS2 stem-loop aptamers (grey) interacting with an MS2 protein tagged with an in vivo biotinylating peptide used to pull down proteins interacting with the RNAs. (**B**) The volcano plot representing proteins captured during MS-based screening showing the magnitude of enrichment (log2 fold change) and the statistical significance (F02Dlog p-value); red dots indicate proteins significantly enriched (p < 0.05) on *FMR1 RNA* compared to GC-rich RNA. Three proteins, RPS26, LUC7L3, and DHX15, tested in a subsequent validation experiment are marked. DDX3X is also indicated as this protein has been previously described in the context of interaction with *FMR1* mRNA. (**C**) The scheme of RNA used for the transient overexpression of FMR99xG (mutant, long, 99 polyglycine tract-containing protein). The construct contains the entire length of the 5'UTR of *FMR1* with expanded CGG repeats forming a hairpin structure (red) tagged with enhanced green fluorescent protein (eGFP) (green). Western blot analysis of FMR99xG and Vinculin for HEK293T cells with insufficient DHX15, RPS26, and LUC7L3 induced by specific short interfering RNA (siRNA) treatment. To detect FMR99xG, the 9FM antibody was used. The upper bands were used for quantification. The graph presents the mean signal for FMR99xG normalized to Vinculin from N = 3 biologically independent samples with the standard deviation (SD). An unpaired Student's t-test was used to calculate statistical significance: *, p < 0.05; ***, p < 0.001; ns, non-significant. (**D**) Results of luciferase assay for cells with overexpression of FMR16xG

*Figure 1 continued on next page*

*Figure 1 continued*

fused with nanoluciferase pretreated with different concentrations of siCtrl or siRPS26 (see legend). The graph presents luminescence signals (means from N=8 with SDs) normalized to siCtrl treated cells. Western blot represents the siRNA-concentration dependent effect on RPS26 downregulation. An unpaired Student's t-test was used to calculate statistical significance: ****, p < 0.0001; ns, non-significant. (**E**) Results of microscopic quantification of FMR99xG-positive aggregates (white arrows) in HeLa cells transfected with the FMR99xG construct on the background of normal (siCtrl) or reduced amount of RPS26 (siRPS26). Representative images were pseudo-colored and merged; green, GFP-positive inclusions; blue, nuclei stained with Hoechst 33342; scale bars, 50 µm. The histogram presents a number of FMRpolyG-GFP inclusions per nucleus in cells treated with siCtrl (blue) or siRPS26 (red) in N=10 biologically independent samples with SDs. *In average, 1100 cells were identified per single image and final value were normalized to 1 for control siRNA treated cells.* An unpaired Student's t-test was used to calculate statistical significance: *, p < 0.05. (**F**) *The* influence of RPS26 silencing on apoptosis evoked in HeLa cells after 28 hr or 43 hr of FMR99xG overexpression or in mock treated cells. Apoptosis was measured as luminescence signals (relative luminescence units; RLU) derived from Annexin V fusion protein bound to phosphatidylserine (PS) exposed on the outer leaflet of cell membranes, an indicator for early apoptosis. The graph presents relative mean values from N = 6 biologically independent samples treated with either siCtrl (red) or siRPS26 (blue) with the SD normalized to mock controls (cells transfected only with the delivering reagent and cultured for 28 hr or 43 hr). *An* unpaired Student's t-test was used to calculate statistical significance: *, p<0.05; **, p<0.01; ***, p < 0.001. Note, that significantly higher apoptosis was observed in cells expressing FMR99xG compare to control (mock), and that pretreatment with siRPS26 significantly reduced this phenotype.

The online version of this article includes the following source data and figure supplement(s) for figure 1:

**Source data 1.** PDF containing tiff files representing western blot results with labeled lanes matching the content of *Figure 1*.

**Source data 2.** Folder containing raw tiff files representing western blot results matching the content of *Figure 1*.

**Figure supplement 1.** MS-based screening revealed proteins interacting with *FMR1 RNA*.

**Figure supplement 1—source data 1.** PDF containing tiff files representing western blot results with labeled lanes matching the content of *Figure 1— figure supplement 1*.

**Figure supplement 1—source data 2.** Folder containing raw tiff files representing western blot results matching the content of *Figure 1—figure supplement 1*.

insufficiency of these proteins on the level of toxic FMRpolyG, which could indicate their role in modulating RAN translation. These experiments were conducted in HEK293T cells transiently expressing FMRpolyG containing 99 glycine residues tagged with GFP, named FMR99xG (*Figure 1C*). Silencing of mRNA encoding for the ATP-dependent RNA helicase DHX15 and RPS26, but not six other analyzed proteins, resulted in a significant decrease in steady-state level of FMR99xG, with no effect on its mRNA (*Figure 1C and A*, *Figure 1—figure supplement 1*). Moreover, using other reporter system expressing polyglycine protein from short CGG repeats (16xCGG) fused with nanoluciferase tag, we showed that partial knock down of RPS26 resulted in significant FMRpolyG downregulation (*Figure 1D*). This indicates that the impact of RPS26 insufficiency on polyglycine level is independent of CGG repeat length, type of reporter used, and that even partial silencing of RPS26 is sufficient to significantly decrease RAN translation from *FMR1* mRNA.

We also investigated whether RPS26 depletion affected the efficiency of FMRpolyG aggregates formation and cell toxicity. In HeLa cells transiently expressing FMR99xG, the frequency of GFP-positive aggregates was reduced upon RPS26 silencing (*Figure 1E*). Moreover, cells with depleted RPS26 exhibited significantly lower apoptosis tendencies evoked by toxic FMRpolyG (*Figure 1F*). These results suggest that decreasing level of RPS26 helps to alleviate FXPAC-related phenotype in cell models.

An orthogonal biochemical assay was used to confirm coprecipitation of RPS26 with the 5'UTR of *FMR1*. We applied an in vitro RNA-protein pull-down using three biotinylated RNAs: 5'UTR of *FMR1* with 99xCGG repeats, synthetic RNA with 23xCGG repeats, and GC-rich RNA as a control. All three RNAs were incubated with an extract derived from HEK293T cells. The first two RNAs, but not GC-rich RNA, were enriched with RPS26, however, the anticipated interaction between RNAs containing CGG repeats and RPS26 was not solely dependent on the triplet number, as RPS26 was pulled down with similar efficiency by RNAs containing more or fewer repeats (*Figure 1A*, *Figure 1—figure supplement 1*).

In sum, we identified 32 proteins enriched on mutant *FMR1 RNA*. RPS26 and DHX15 insufficiency hindered FMR99xG RAN translation efficiency in a transient expression system. Notably, among proteins identified in the screening there were the ATP-dependent RNA helicase (DDX3X)—the protein previously described as RAN translation modifier (*Linsalata et al., 2019*), and the Src-associated in mitosis 68 kDa protein (SAM68)—the splicing factor sequestered on RNA containing CGGexp (*Sellier et al., 2010*; *Figure 1B and A*, *Figure 1—figure supplement 1*). Identified interactors can be involved

not only in RAN translation, but potentially also in different metabolic processes of mutant *FMR1* RNA, such as transcription, protein sequestration, RNA transport, localization, and stability. Importantly, our data showed that silencing of RPS26 alleviated the pathogenic effect of toxic FMRpolyG produced from *FMR1* mRNA containing CGGexp. These facts encouraged us to further investigate the role of RPS26 in the context of CGG-exp-related RAN translation in more natural models.

## RPS26 depletion affects cells proliferation but does not inhibit global translation

The RPS26 is a component of the 40 S ribosomal subunit and its inclusion or depletion from this subunit affects translation of selected mRNA but does not impact overall translation rate (*Schütz et al., 2018*; *Ferretti et al., 2017*; *Havkin-Solomon et al., 2023*; *Yang and Karbstein, 2022*; *Gaikwad et al., 2021*). For instance, in yeast cultured in stress conditions, Rps26 is disassociated from 40 S subunit, which results in translation of different mRNA pool, especially those mRNAs encoding proteins implicated in stress response pathway (*Ferretti et al., 2017*; *Yang and Karbstein, 2022*). Moreover, it was shown that RPS26 is involved in translational regulation of selected mRNA contributing to the maintenance of pluripotency in murine embryonic stem cells (*Li et al., 2022*). Its C-terminal domain interacts with mRNA sequences upstream to the E-site of an actively translating ribosome (*Pisarev et al., 2008*; *Hussain et al., 2014*; *Anger et al., 2013*). This localization may be responsible for the regulation of efficient translation initiation (*Pisarev et al., 2008*; *Havkin-Solomon et al., 2023*), especially in a context of non-canonical RAN translation. Hence, these facts encouraged us to further investigate this protein in the context of CGGexp-related RAN translation.

It was recently demonstrated that RPS26 insufficiency affects cells viability *Havkin-Solomon et al., 2023*, hence we asked if RPS26 silencing affects the proliferation rate of HEK293T cells. Forty-eight hours post RPS26 silencing, cells divided less frequently as their number was reduced by 34% in comparison to control siRNA-treated cells (*Figure 2A*). Importantly, almost no dead cells were detected indicating that RPS26 depletion is not detrimental for the cells (*Figure 2A*). Given that RPS26 is crucial for 40 S maturation *Plassart et al., 2021*, we verified if RPS26 depletion affects global translation. Surface sensing of translation (SUnSET) assay *Schmidt et al., 2009* demonstrated that RPS26 insufficiency did not elicit significantly the global translation inhibition, as opposed to cycloheximide (CHX) treatment used as a positive control (*Figure 2B*).

## RPS26 acts as a RAN translation modulator of mRNA with short and long CGG repeats

To monitor FMRpolyG RAN translation efficiency and test the modulatory properties of preselected proteins, we generated two cell lines stably expressing a fragment of an *FMR1* gene with expanded (95xCGG) or short, normal CGG repeats (16xCGG), named *S-95xCGG* and *S-16xCGG*, respectively. These two models were generated by taking advantage of the Flp-In T-Rex 293 system (*Szczesny et al., 2018*), which expresses transgenes encoding for either longer (FMR95xG) or shorter (FMR16xG) polyglycine tract-containing proteins tagged with EGFP under the control of a doxycycline-inducible promoter (*Figure 3A*). In these cell lines, the expression of transgenes is detectable a few hours after promoter induction (*Figure 3—figure supplement 1*); however, even in prolonged doxycycline exposure (up to 35 days), we did not observe FMR95xG positive aggregates (*Figure 3—figure supplement 1*). A lack of aggregation is advantageous for monitoring RAN translation efficiency, as it allows the entire pool of RAN proteins present in cellular lysate to be measured by western blot, which is impossible if part of the proteins is trapped in insoluble aggregates (*Derbis et al., 2021*; *Derbis et al., 2018*). Moreover, single copy integration of the transgene containing a fragment of the *FMR1* gene mimics natural situation. Notably, the level of transgene expression was comparable to *FMR1* endogenous expression (*Figure 3—figure supplement 1*) and was homogeneous between cells (*Figure 3A*, *Figure 3—figure supplement 1*). The second model, named *L-99xCGG*, was generated using lentiviral transduction of HEK293T cells (*Figure 3D*). Lentiviral particles were prepared based on genetic construct used in the transient transfection system (*Figure 1C*). This stable cell line constantly expresses FMR99xG tagged with GFP. In the *L-99xCGG* model, FMR99xG was present in soluble and, to a lesser extent, aggregated form (*Figure 3D*, *Figure 3—figure supplement 1*).

siRNA-induced silencing of RPS26 in the stable *S-95xCGG* cell model expressing mRNA with long 95xCGG repeats resulted in a significantly decreased level of steady-state FMR95xG, with no

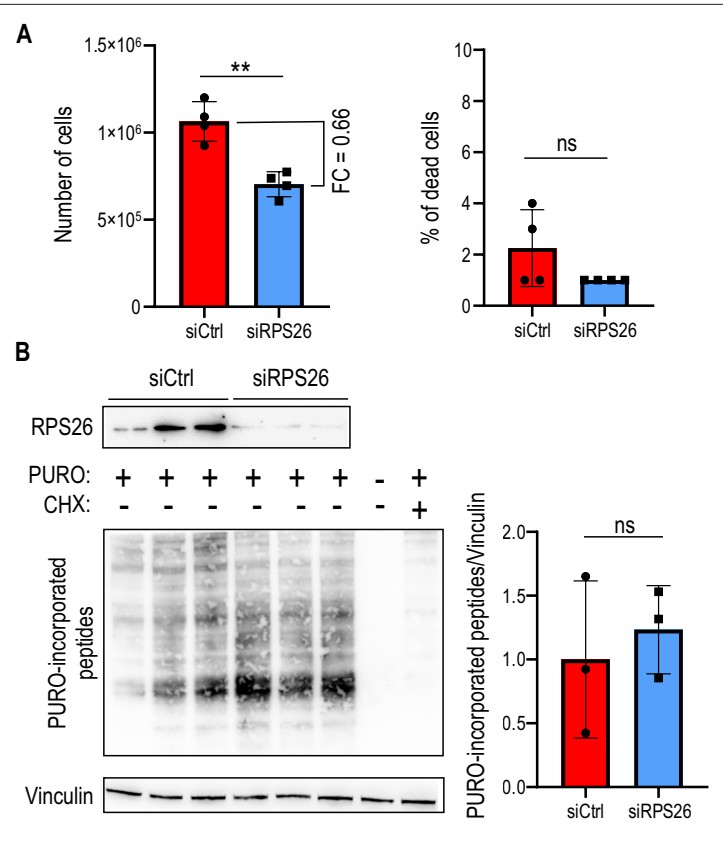

**Figure 2.** RPS26 depletion affects proliferation of cells but does not inhibit global translation. (**A**) The left graph represents the number of cells and right graph represents percentage (%) of dead cells positively stained with trypan blue after 48 hr post siRNA-induced silencing. Prior siRNA transfection equal amount of HEK293T cells were seeded (ca. 1x10⁴). The graphs present means from N = 4 with SDs. FC, fold change. An unpaired Student's t-test was used to calculate statistical significance: **, p<0.01; ns, non-significant. Note, that the number of cells in samples treated with siRPS26 was lowered by 34% compared to siCtrl treated samples. (**B**) Western blot analysis of the surface sensing of translation (SUnSET) assay 48 hr post RPS26 silencing (siRPS26) in HEK293T cells, with antibody against puromycin (PURO), to monitor efficiency of PURO incorporation into peptides during translation. Cells were incubated with siRPS26 or siCtrl and 10 min before protein isolation PURO was added. As a positive control for efficient translation inhibition cells were treated with cycloheximide (CHX), an inhibitor of translation elongation (CHX was added 5 min prior PURO treatment). As a negative control cells treated with DMSO only were used (negative for both PURO and CHX). The graph represents mean signal from whole lanes from N=3 independent samples with SDs. An unpaired Student's t-test was used to calculate statistical significance: ns, non-significant.

The online version of this article includes the following source data for figure 2:

**Source data 1.** PDF containing tiff files representing western blot results with labeled lanes matching the content of *Figure 2*.

**Source data 2.** Folder containing raw tiff files representing western blot results matching the content of *Figure 2*.

---

effect on its mRNA (*Figure 3B*, *Figure 3—figure supplement 1*). Moreover, the RPS26 silencing in the *S-16xCGG* model also decreased the level of RAN protein product derived from short 16xCGG repeats without affecting its mRNA level (*Figure 3C*). This suggests that RPS26 depletion affects RAN protein levels independently of CGG repeat content. Similarly to the *S-95/16xCGG* models, RPS26 silencing in the lentivirus integration-based *L-99xCGG* model also significantly downregulated FMR99xG biosynthesis (*Figure 3E*).

Previously, several proteins known as RPS26 responders, for example murine Polycomb protein (Suz12) and Histone H3.3, were shown to be negatively affected by RPS26 depletion (*Li et al., 2022*). Here, we showed that they also negatively responded to the RPS26-specific siRNAs used in this study (*Figure 3—figure supplement 1*).

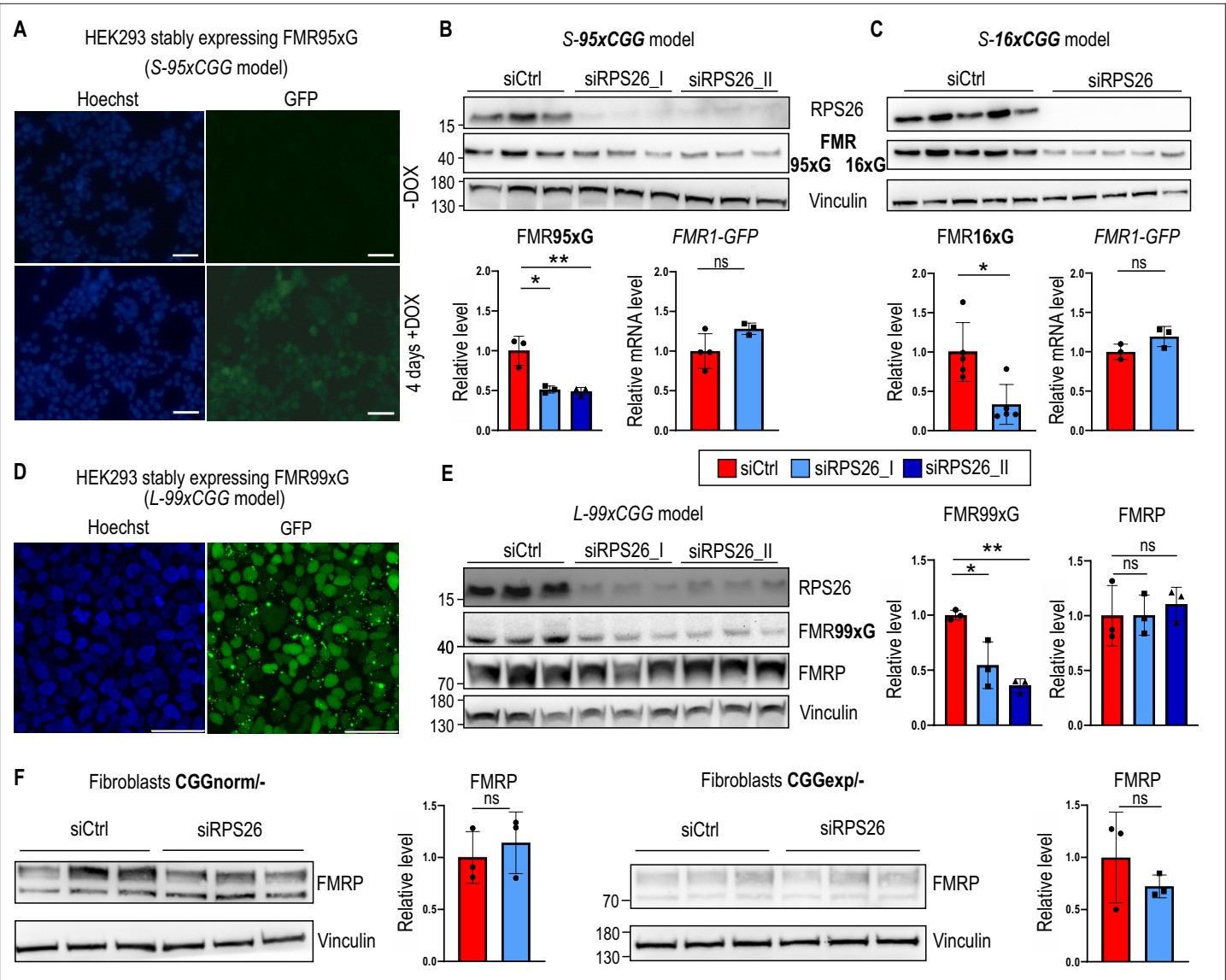

**Figure 3.** RPS26 insufficiency induces a lower production of polyglycine, but not FMRP, in multiple FXTAS cellular models. (**A**) Representative microscopic images showing inducible expression of FMR95xG fused with eGFP in stable transgenic cell line containing a single copy of 5'UTR FMR1 99xCGG-eGFP transgene under the control of a doxycycline-inducible promoter: *S-95xCGG model*. Blue are nuclei stained with Hoechst; the green signal is derived from FMR95xG tagged with eGFP; scale bar 50 μm;+DOX and F02DDOX indicate cells treated (or not) with doxycycline to induce transcription of the transgene from the doxycycline-dependent promoter. (**B & C**) Results of western blot analyses of FMR95xG (with long, mutant polyglycine stretches) or FMR16xG (with short, normal polyglycine stretches), both fused with GFP, normalized to Vinculin and an RT-qPCR analysis of *FMR1-GFP* transgene expression normalized to *GAPDH* upon RPS26 silencing in *S-95xCGG* and *S-16xCGG* models, respectively. siRPS26_I and siRPS26_II indicate two different siRNAs used for RPS26 silencing. The graphs present means from N = 3 (**B**) or N=5 biologically independent samples (**C**) with SDs. An unpaired Student's t-test was used to calculate statistical significance: *, p<0.05; **, p<0.01; ns, non-significant. (**D**) Representative microscopic images of cells stably expressing FMR99xCGG obtained after transduction with lentivirus containing the 5'UTR FMR1 99xCGG-eGFP transgene: *L-99xCGG* model. Nuclei are stained with Hoechst and green fluorescence signal is derived from FMR99xG tagged with GFP. Scale bar 50 μm. (**E**) Results of western blot analyses of FMR99xG and FMRP normalized to Vinculin upon RPS26 silencing in the *L-99xCGG* model. The graphs present means from N = 3 biologically independent samples with SDs. An unpaired Student's t-test was used to calculate statistical significance: *, p<0.05; **, p<0.01; ns, non-significant. (**F**) Results of western blot analyses of FMRP levels normalized to Vinculin upon RPS26 silencing in fibroblasts derived from two males, either from healthy individual (CGGnorm – XY, 31xCGG) or FXTAS patient (CGGexp – XY, 81xCGG). The graphs present means from N = 3 biologically independent samples with SDs. An unpaired Student's t-test was used to calculate statistical significance: ns, non-significant.

The online version of this article includes the following source data and figure supplement(s) for figure 3:

**Source data 1.** PDF containing tiff files representing western blot results with labeled lanes matching the content of *Figure 3*.

**Source data 2.** Folder containing raw tiff files representing western blot results matching the content of *Figure 3*.

*Figure 3 continued on next page*

*Figure 3 continued*

**Figure supplement 1.** Characterization of *S-95xCGG* and *S-16xCGG* models and RPS26-responders.

**Figure supplement 1—source data 1.** PDF containing tiff files representing western blot results with labeled lanes matching the content of *Figure 3—figure supplement 1*.

**Figure supplement 1—source data 2.** Folder containing raw tiff files representing western blot results matching the content of *Figure 3—figure supplement 1*.

We further investigated whether the translation of FMRP might be affected by RPS26 depletion, as an open reading frame for FMRpolyG–which appears to be under the control of RPS26-sensitive translation–is derived from the same mRNA. To test this hypothesis, we treated human fibroblasts derived from a FXTAS patient and a healthy control with RPS26-specific siRNA. Our results indicate that RPS26 depletion does not affect FMRP levels, regardless of CGG repeat length in *FMR1* mRNA (*Figure 3F*). Similarly, the FMRP level was not affected by RPS26 depletion in other cell models (*Figure 3E*).

Together, these results imply that although the presence of RPS26 in 40 S subunit has a positive effect on RAN translation of FMRpolyG and other previously identified RPS26 responders, it does not affect the canonical translation of FMRP produced from the same mRNA. Moreover, a lack of significant differences in translation efficiency of long and short polyglycine-containing proteins in cells with insufficiency of RPS26 suggests that observed effect may depend on certain RNA sequences or structures within the 5'UTR of *FMR1* mRNA or other features of this mRNA, rather than the CGG repeat length.

## RPS26 depletion affects only a small subset of the human proteome

To assess globally which proteins are sensitive to RPS26 insufficiency, we used stable isotope labeling with amino acids in cell culture (SILAC). The protein expression level in control and RPS26-depleted HEK293T cells was determined via quantitative mass spectrometry (SILAC-MS, *Supplementary file 1, table 6*). Using this approach, we identified over 2600 proteins, with more than 80% identifications based on at least two unique peptides. Differential data analysis of quantified proteins (N=1887) indicated that most (ca. 80%) proteins identified by SILAC-MS were not sensitive to RPS26 deficiency; we named these proteins non-responders (N=1506; *Supplementary file 1, table 7*). We also identified set of proteins that negatively (negative responders; N=223 if the p-value was <0.05) or positively responded (positive responders; N=158 if the p-value <0.05) to RPS26 deficiency (*Figure 4A*, *Supplementary file 1, table 7*). Non-responders were used as a background list in the GO analysis. An analysis of positive responders found that the proteins in this group were mainly components of translation initiation complexes or formed large ribosomal subunits (*Figure 4B*, *Supplementary file 1,table 8*). A similar analysis for negative responders did not reveal any significantly enriched GO terms (*Supplementary file 1, table 8*), partially due to the small number of identified proteins. To further validate this data, we performed western blots for selected negative responders (PDCD4, ILF3, RPS6, and PCBP2), non-responders (FMRP and FUS) or positive responders (EIF5 and EIF3J) from independent cell samples collected 48 hr post siRPS26 treatment (*Figure 4C*, *Figure 4—figure supplement 1*). Most tested proteins (6 out of 8) aligned with the quantitative SILAC-MS data (*Figure 4C*, *Figure 4—figure supplement 1*).

Previously published results showed that specialized ribosomes are formed to selectively translate mRNAs with specific features (*Genuth and Barna, 2018*; *Shi et al., 2017*). In yeast, mRNAs translated by Rps26-depleted ribosomes lack conservation of all Kozak sequence elements, while mRNAs translated by Rps26-containing ribosomes present a full Kozak consensus (*Ferretti et al., 2017*). RPS26 is localized next to the E-site of the translating ribosome and was found to interact with template mRNAs (*Pisarev et al., 2008*; *Anger et al., 2013*). For instance, if the 43 S recognizes the start codon, the RPS26 contacts position −4 from the start codon in yeast (*Ferretti et al., 2017*) and from −11 up to −16 of attached mRNA in mammals (*Havkin-Solomon et al., 2023*). Given that RPS26 may be involved in start codon fidelity (*Ferretti et al., 2017*; *Havkin-Solomon et al., 2023*), we searched for specific sequence motifs in mRNAs encoding proteins responding to RPS26 deficiency. We investigated sequences containing 5'UTRs and coding sequences (CDSs) close to the start codon of mRNAs encoding positive and negative RPS26 responders. We used the total human transcriptome, named background (BG; N=22,160), as a reference. We did not observe any significant differences in the frequency of individual nucleotide positions in the 20-nucleotide vicinity of the start codon relative to

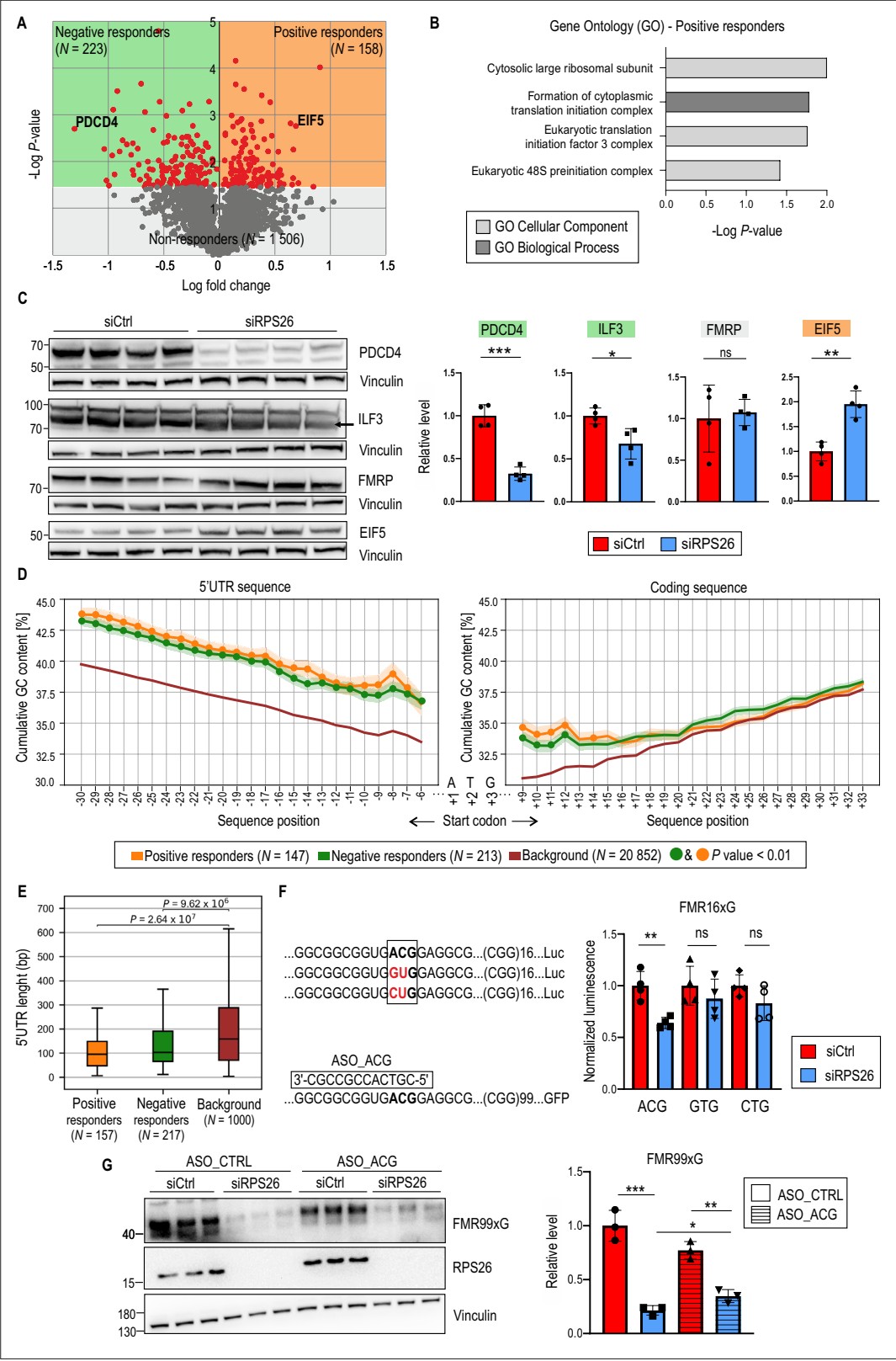

**Figure 4.** Changes in the proteome of RPS26-deficient cells are not robust. (**A**) The volcano plot represents a stable isotope labeling using amino acids in cell culture (SILAC)-based quantitative proteomic analysis identifying proteins sensitive to RPS26 insufficiency. It shows the magnitude of protein-level changes (log2 fold change) vs. the statistical significance (F02Dlog p-value) 48 hr post siRPS26 treatment. Data were collected from

*Figure 4 continued on next page*

*Figure 4 continued*

three independent biological replicates for each group. Grey dots indicate proteins non-responding to RPS26 depletion (N=1506). Red dots indicate proteins responding to RPS26 insufficiency (p<0.05); EIF5 and PDCD4 are examples of Negative (N=223; green) and Positive (N=158; orange) responders, respectively. Protein groups were further analyzed in B, C, and D. (**B**) Gene ontology (GO) analysis performed for positive responders of RPS26 insufficiency from the proteomic experiment shown in A. The graph presents significantly enriched GO terms (p<0.05); statistical significance was calculated using Fisher's Exact test with the Bonferroni correction. Note that for negative responders, no GO terms were significantly enriched. (**C**) Data validation from the proteomic experiment described in A. Western blot analyses of PDCD4, ILF3, FMRP, and EIF5 proteins normalized to Vinculin for HEK293T cells treated with siRPS26. The graphs present means from N = 4 biologically independent samples *with SDs.* An unpaired Student's t-test was used to calculate statistical significance: *, p<0.05; **, p<0.01; ***, p < 0.001; ns, non-significant. (**D**) The percentage of GC content in 5'UTRs and coding sequences across extending sequence windows initiated from the start codon within three groups of transcripts: Negative responders (green), Positive responders (orange), and Background (BG) understood as the total transcriptome (red). To avoid biases in the GC-content mean, transcripts with 5'UTR sequences shorter than 20 nucleotides were excluded from the analysis, yielding the following sample sizes: Negative responders (N=213), Positive responders (N=147), and BG (N=20,862). For example, position F02D6 in 5' UTR sequences corresponds to a 6-nucleotide fragment (window from F02D6 to F02D1 positions upstream of ATG), while position F02D7 corresponds to a 7-nucleotide fragment (window from F02D7 to F02D1 positions). The solid line shows the mean GC content at a given position (i.e., within the window), and the shade indicates the standard error of the mean. *P*-values <0.01 are denoted by green or yellow dots. These reflect pairwise comparisons of GC content between transcript groups (compared to BG) and were determined using a two-tailed paired t-test with Bonferroni correction. (**E**) Box plot showing the results of a one-tailed Mann-Whitney U test comparing 5'UTR sequence lengths between positive and negative responders relative to the background transcriptome. To ensure a balanced comparison, the background transcriptome is represented by 1000 randomly selected transcripts from the complete human transcriptome. The plot highlights the median (indicated by the central line), the upper and lower quartiles (represented by the boxes), and the highest and lowest values, which are determined within a range of 1.5 times the interquartile range (IQR) (whiskers). (**F**) Results of luciferase assay post RPS26 silencing showing differences in expression of transgene containing 5'UTR of *FMR1* with 16xCGG fused with nanoluciferase from which RAN translation was initiated from either ACG (wild type) or GTG or CTG near cognate codons. The signals for activity of nanoluciferase fused with FMR16xG were normalized to firefly luciferase signals (means from N=4 with SDs are presented on the graph). Note that firefly luciferase was translated from canonical ATG start codon. An unpaired Student's t-test was used to calculate statistical significance: **, p<0.01; ns, non-significant. (**G**) Western blot analysis of FMR99xG post RPS26 silencing and treatment with control ASO (ASO_CTRL) or ASO targeting ACG codon, RAN translation initiation codon in 5'UTR of FMR1 (ASO_ACG). The graph represents the means of N=3 with SDs. An unpaired Student's t-test was used to calculate statistical significance: *, p<0.05; **, p<0.01; ***, p < 0.001; ns, non-significant.

The online version of this article includes the following source data and figure supplement(s) for figure 4:

**Source data 1.** PDF containing tiff files representing western blot results with labeled lanes matching the content of *Figure 4*.

**Source data 2.** Folder containing raw tiff files representing western blot results matching the content of *Figure 4*.

**Figure supplement 1.** Validation of SILAC-MS results and bioinformatic analysis of mRNAs encoding RPS26-sensitive proteins.

**Figure supplement 1—source data 1.** PDF containing tiff files representing western blot results with labeled lanes matching the content of *Figure 4—figure supplement 1*.

**Figure supplement 1—source data 2.** Folder containing raw tiff files representing western blot results matching the content of *Figure 4—figure supplement 1*.

the expected distribution in the BG (*Figure 4—figure supplement 1*, *Supplementary file 1, table 9*). However, we identified a significantly higher GC content in the 5'UTR sequences in the positive and negative responder groups relative to the BG at all analyzed positions from −6 to −30 (*Figure 4D*). Although the difference in the GC content in CDSs of responders relative to the BG was much smaller than for the 5'UTRs, we observed a significantly higher (p<0.05) GC content in the close vicinity of the start AUG codon at upstream positions from +9 up to +14 (*Figure 3D*). Moreover, when we applied more stringent selection criteria (p<0.01), the number of analyzed transcripts was significantly reduced (positive responders; N=42, negative responders; N=54), but the GC richness appeared to be more predominant for mRNAs encoding negative RPS26 responders (*Figure 4—figure supplement 1*). Additionally, we found that 5'UTRs of proteins encoding positive and negative RPS26 responders

were significantly shorter in comparison to reference represented by 1000 randomly selected transcripts from the complete human transcriptome (*Figure 4E*).

Although bioinformatic analyses did not reveal any importance of position −4 from the start codon in any group of RPS26 responders (*Figure 4—figure supplement 1*), we experimentally verified whether changes in this position in human cells would affect RAN translation initiation. Using mutagenesis, we substituted G to A in the −4 position from ACG near cognate codon of *FMR1*, as A has previously been shown to be an enhancer of translation initiation in yeast 43 S containing Rps26 (*Ferretti et al., 2017*; *Ferretti et al., 2018*). We did not observe significant differences in efficiency of FMRpolyG biosynthesis between the two tested mRNAs in human HEK293T cells, confirming our bioinformatic data (*Figure 4—figure supplement 1*). Additionally, to verify if some of the near-cognate codons are more sensitive to RPS26 depletion, we substituted ACG for GTG or CTG codons in 5'UTR of *FMR1* with 16xCGG in frame with nanoluciferase reporter. We found that RPS26 silencing influenced only the level of FMRpolyG initiated at ACG near cognate start codon (*Figure 4F*). To further investigate importance of codon selection by RPS26, we simultaneously knocked down RPS26 and delivered short antisense oligonucleotides (ASO) blocker targeting ACG codon to cells expressing FMRpolyG. We found that the effect of RPS26 depletion on FMR99xG synthesis was hindered by ASO treatment and the depth of the effect is different than in siCtrl treated cells (*Figure 4G*), suggesting that accessibility of ACG start codon in *FMR1* mRNA is required for RPS26 level-sensitive translation.

We also searched for specific, short sequence motifs, *k*-mers, which might have been enriched in the 5'UTR of mRNAs encoding responders to RPS26 deficiency. We identified a list of 6 and 14 significantly over-represented hexamers ($p<10^{-5}$) in positive and negative responder groups, respectively. These predominantly comprised G(s) and C(s). For example, the most over-represented hexamers in the 5'UTRs of the positive responders ($p<10^{-10}$) included GCCGCC, CCGCTG, and CCGGTC, and for negative responders, the highest over-representation ($p<10^{-6}$) had CGCCGC, GCCGCC, and GCGGCG (*Figure 4—figure supplement 1*, *Supplementary file 1, table 10*).

Altogether, these data indicate that RPS26 depletion may impact the translation rate of mRNAs containing GC-rich sequences in shorter 5'UTRs and in close proximity downstream to the AUG start codon (up to 14 nucleotides). It may suggest that thermodynamic stability of RNA structure formed upstream or downstream of a scanning 43 S PIC and perhaps the dynamics of translocation of this complex could be a factor modulating sensitivity to RPS26 insufficiency. We did not see any importance of A or G in −4 position from ACG initiation codon on FMRpolyG biosynthesis in cells having or lacking RPS26; however, we provided the evidence that cells with downregulated RPS26 display near-cognate start codon selection regulatory properties. Our data showed also that the translation of most human proteins, including FMRP, was unaffected by the impairment of RPS26 level.

## TSR2 mediates the RAN translation of FMRpolyG, perhaps via RPS26

The *pre-rRNA-processing protein TSR2* (*Tsr2*) is a chaperone-acting protein that regulates the Rps26 cellular level and is responsible for incorporating Rps26 into 90 S pre-ribosome in yeast nuclei (*Schütz et al., 2014*; *Schütz et al., 2018*). RPS26 inclusion into pre-40S together with other factors facilitates final step of 40 S subunit maturation (*Plassart et al., 2021*). Moreover, Tsr2 mediates the disassembly of Rps26 from mature 40 S subunit in cytoplasm in high salt and pH conditions (*Yang and Karbstein, 2022*) or when Rps26 is oxidized (*Yang et al., 2023*).

Assuming that depletion of TSR2 in mammalian cells may affect the level of RPS26 loaded on 40 S in human cells, we hypothesized that silencing TSR2 would affect the biosynthesis of FMRpolyG. Indeed, in stable *S-95xCGG model* (*Figure 5A*) and COS7 cell line (*Figure 5—figure supplement 1*) treated with siRNA against TSR2 (siTSR2) we observed lower level of both RPS26, FMR95xG or transiently expressed FMR99xG, respectively, but the level of FMRP was unchanged (*Figure 5A*). The effect of TSR2 silencing on RAN translation was independent of CGG repeat length, as similar results were obtained for smaller FMR16xG proteins (*Figure 5—figure supplement 1*). Silencing TSR2 did not affect the level of other components of the 40 S subunits such as RACK1, RPS6 and RPS15 however, the level of a known responder of RPS26 insufficiency, Histone H3.3 (*Li et al., 2022*), was significantly reduced (*Figure 5A&B*). The observed reduction of FMR95xG and FMR16xG protein by TSR2 depletion did not stem from global translation inhibition, as TSR2 silencing did not impact the overall translation rate (*Figure 5C*). Moreover, to confirm selective impact of RPS26 on polyglycine biosynthesis, we tested the effect of silencing of other small ribosomal subunit protein, RPS6, on RAN translation

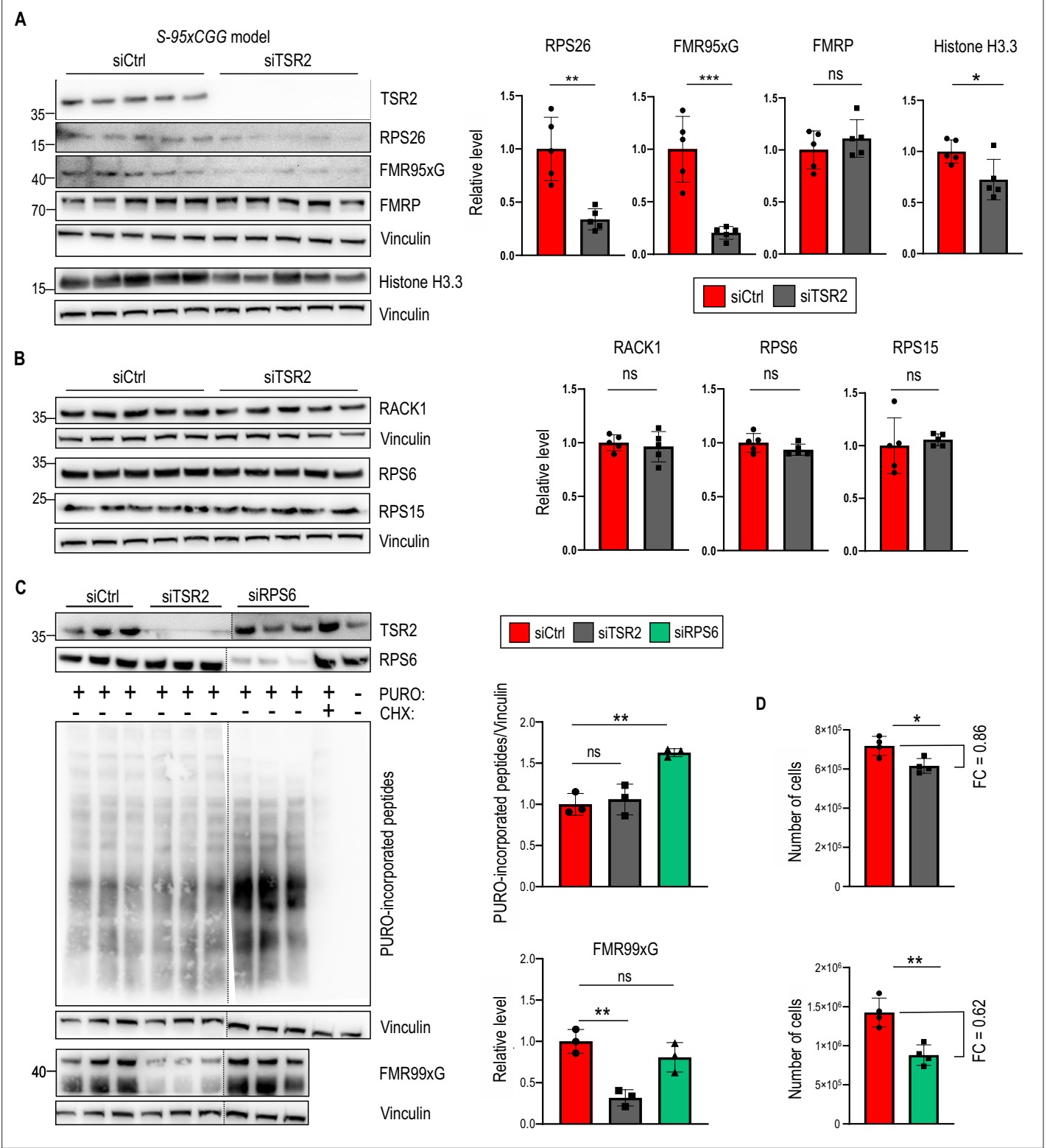

**Figure 5.** Insufficiency of the TSR2 chaperone protein lowers the level of RPS26 and FMRpolyG but not FMRP and selected 40 S components. (**A**) Western blot analyses of RPS26, FMR95xG, FMRP but also Histone H3.3, a sensor of ribosomes depleted with RPS26, normalized to Vinculin in stable *S-95xCGG* cells treated with siTSR2. Graphs represent means from N = 5 biologically independent samples *with SDs*. An unpaired Student's t-test was used to calculate statistical significance: **, p<0.01; ***, p < 0.001; ns, non-significant. (**B**) Western blot analyses of selected 40 S ribosomal proteins RACK1, RPS6, and RPS15, normalized to Vinculin upon TSR2 silencing in the *S-95xCGG* model. The graphs present means from N = 5 biologically

*Figure 5 continued on next page*

*Figure 5 continued*

independent samples *with SDs*. An unpaired Student's t-test was used to calculate statistical significance: *, p<0.05; ns, non-significant. (**C**) Western blot analysis of the results of SUnSET assay (upper) and the level of FMR99xG (lower) 48 hr post TSR2 or RPS6 silencing in HEK293T cells. CHX treatment served as a positive control for translation inhibition. Negative control (PURO negative and CHX negative) was protein extract from cells treated with DMSO only (see *Figure 2* for more details). Graphs represent means form N=3 independent samples with SDs. An unpaired Student's t-test was used to calculate statistical significance: **, p<0.01; ns, non-significant. Note that presented blots are cropped. Full size images are deposited on Zenodo server under the DOI: 10.5281/zenodo.13860369. (**D**) Graphs represent the number of cells post 48 hr of TSR2 or RPS6 silencing. Prior siRNA transfection equal amount of HEK293T cells were seeded (~1 x 10$^4$). The graphs present means from N = 4 with SDs. FC, fold change. An unpaired Student's t-test was used to calculate statistical significance: *, p<0.05; **, p<0.01.

The online version of this article includes the following source data and figure supplement(s) for figure 5:

**Source data 1.** PDF containing tiff files representing western blot results with labeled lanes matching the content of *Figure 5*.

**Source data 2.** Folder containing raw tiff files representing western blot results matching the content of *Figure 5*.

**Figure supplement 1.** Silencing of TSR2 reduces FMR99xG and FMR16xG level.

**Figure supplement 1—source data 1.** PDF containing tiff files representing western blot results with labeled lanes matching the content of *Figure 5— figure supplement 1*.

**Figure supplement 1—source data 2.** Folder containing raw tiff files representing western blot results matching the content of *Figure 5—figure supplement 1*.

efficiency and found, that RPS6 depletion did not impact FMR99xG level (*Figure 5C*). Additionally, we verified how TSR2 and RPS6 depletion affects cell proliferation rate. Upon TSR2 knockdown, cell number decreased just by around 15% when comparing to control, but RPS6 insufficiency induced decrease of cell number by ca. 50% (*Figure 5D*).

Obtained results could be explained by the fact that insufficiency of TSR2 may affect the incorporation of RPS26 during nuclear maturation (*Schütz et al., 2018*) or cytoplasmic regeneration of 40 S (*Yang et al., 2023*), thus having a negative effect on FMRpolyG biosynthesis, without affecting the global translation. Overall, this data strengthens our conclusion regarding the positive and selective effect of RPS26 and TSR2 on RAN translation initiated from the near-cognate ACG codons located in the 5'UTR of *FMR1*.

## The other 40S subunit component, RPS25, affects CGGexp-related RAN translation

Considering data concerning heterogenous ribosomes and their diverse roles in translation regulation (*Genuth and Barna, 2018*; *Shi et al., 2017*), we hypothesized that other component of the 40 S subunit, the ribosomal protein RPS25, which localizes near RPS26 on 40 S structure (*Pisarev et al., 2008*; *Anger et al., 2013*), may regulate CGGexp-related RAN translation. Importantly, this protein was already identified as a modifier of RAN translation of mRNAs containing other types of expanded repeats–GGGGCCexp and CAGexp (*Yamada et al., 2019*).

Upon silencing of RPS25 in the stable *S-95xCGG* model, we observed significant decline of FMRpolyG level with no effect on its encoding mRNAs (*Figure 6A*). However, we found that RPS25 silencing, like RPS26 silencing, negatively affected HEK293T cells proliferation (*Figure 6B*). Therefore, we also performed similar experiment in human neuroblastoma SH-SY5Y cells, which viability was moderately affected by silencing of all investigated factors (*Figure 6C*). We demonstrated that in SH-SY5Y with transient expression of FMR99xG, the reduction of RAN translation product was comparable after depletion of RPS25 or RPS26 (*Figure 6D*, *Figure 6—figure supplement 1*). We also found that RPS25 coprecipitates with the 5'UTR of *FMR1* that lacks or harbors expanded 99xCGG repeats via biochemical assay using the biotinylated RNA-protein pull-down assay (*Figure 6—figure supplement 1*). Altogether, these results suggest that insufficiency of RPS25, like RPS26, is a factor which negatively affects CGGexp-related RAN translation in polyglycine frame.

## Discussion

RAN translation was first described in 2011 and reported for Spinocerebellar Ataxia type 8 (SCA8) linked to CAG triplet expansion (*Zu et al., 2011*) and in other repeat expansion-related disorders (REDs), such as Huntington's disease (HD; *Bañez-Coronel et al., 2015*), C9orf72-linked amyotrophic

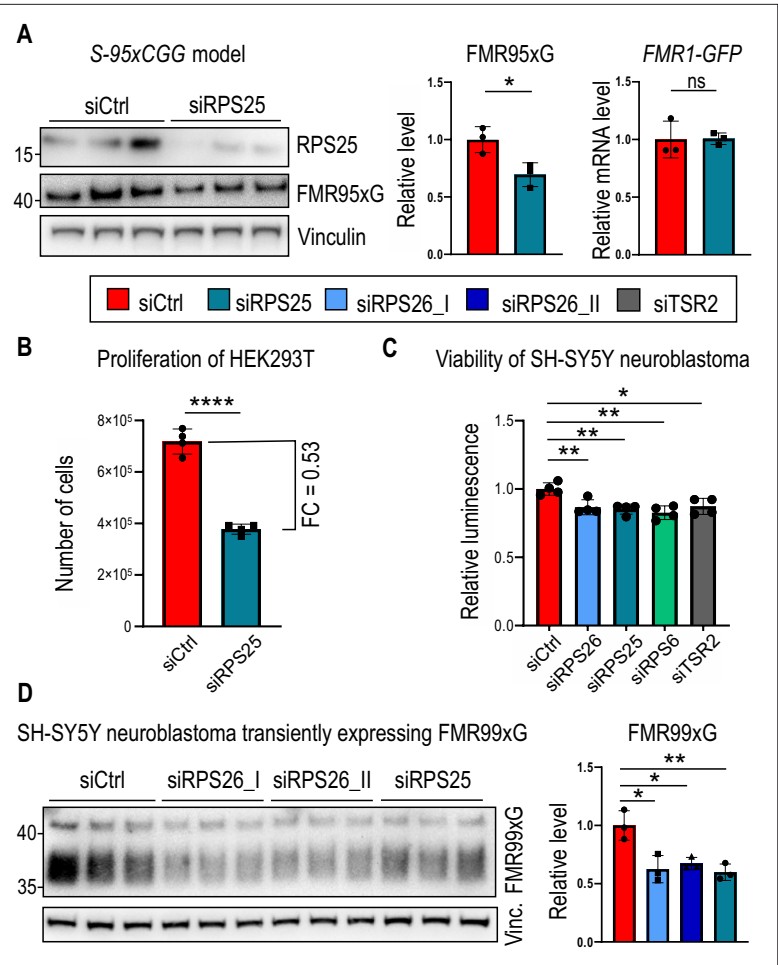

**Figure 6.** Silencing RPS25, the other component of 43 S PIC, reduces the level of polyglycine produced from mutant *FMR1* mRNA. (**A**) A western blot analysis of FMR95xG normalized to Vinculin and an RT-qPCR analysis of the *FMR1-GFP* transgene expression normalized to *GAPDH* upon RPS25 silencing in the *S-95xCGG* model. The graphs present means from N = 3 biologically independent samples *with SDs*. An unpaired Student's t-test was used to calculate statistical significance: *, p<0.05; ns, non-significant. (**B**) The graph represents the number of cells post 48 h of RPS25 silencing. Prior siRNA transfection equal amount of HEK293T cells were seeded (~1 x 10⁴). The graph presents means from N = 4 with SDs. FC, fold change. An unpaired Student's t-test was used to calculate statistical significance: ****, p<0.0001. (**C**) The graph represents viability of SH-SY5Y neuroblastoma cells measured by luminescence based on the reducing potential of viable cells. SH-SY5Y cells were treated with all siRNAs tested in this study. The graph presents means from N = 4 biologically independent samples with SDs. An unpaired Student's t-test was used to calculate statistical significance: *, p<0.05; **, p<0.01. (**D**) A western blot analysis of FMR99xG normalized to Vinculin from the human neuroblastoma cell line (SH-SY5Y) upon silencing of either RPS26 or RPS25. siRPS26_I and siRPS26_II indicate two different siRNAs used for RPS26 silencing. The upper bands were used for quantification. The graph presents means from N = 3 biologically independent samples *with SDs*. An unpaired Student's t-test was used to calculate statistical significance: *, p<0.05; **, p<0.01; ns, non-significant.

The online version of this article includes the following source data and figure supplement(s) for figure 6:

**Source data 1.** PDF containing tiff files representing western blot results with labeled lanes matching the content of *Figure 6*.

**Source data 2.** Folder containing raw tiff files representing western blot results matching the content of *Figure 6*.

**Figure supplement 1.** RPS25 co-precipitates with *FMR1* and affects FMR99xG level.

**Figure supplement 1—source data 1.** PDF containing tiff files representing western blot results with labeled lanes matching the content of *Figure 6—figure supplement 1*.

**Figure supplement 1—source data 2.** Folder containing raw tiff files representing western blot results matching the content of *Figure 6—figure supplement 1*.

lateral sclerosis (C9-ALS), and frontotemporal dementia (FTD; *Mori et al., 2013*; *Ash et al., 2013*), which correlate to CAG and GGGGCC repeats expansions, respectively. RAN translation contribute to the development and progression of many REDs (*Zu et al., 2011*; *Mori et al., 2013*; *Ash et al., 2013*; *Bañez-Coronel et al., 2015*), including FXTAS (*Todd et al., 2013*; *Sellier et al., 2017*) and FXPOI (*Buijsen et al., 2016*); however, no effective therapy targets this pathomechanism. Currently, some RAN translation modifiers of *FMR1* mRNA have been described. For instance, RNA helicases such as DDX3X (*Linsalata et al., 2019*) or DHX36 (*Tseng et al., 2021*), and others extensively reviewed in *Baud et al., 2022* have been identified as affecting CGG repeats-related RAN translation by facilitating ribosomal scanning via unwinding the structured RNA. Components of the endoplasmic reticulum ER-resident kinase (PERK) signaling pathway were shown to modulate RAN translation under stress conditions (*Green et al., 2017*), and the activity of another kinase, SRPK1, retains mutated *FMR1* mRNA containing CGGexp in the nucleus blocking its transport to the cytoplasm and subsequent translation (*Malik et al., 2021a*). Recently, it was demonstrated that RAN translation activates ribosome-associated quality control (RQC) pathway, which prevents accumulation of RAN misfolded proteins by ubiquitination and subsequent proteasomal degradation (*Tseng et al., 2024*). Nevertheless, mechanistic insights into RAN translation remain elusive.

Our research aimed to reveal proteins coprecipitating with the 5'UTR of *FMR1* mRNA containing expanded CGG repeats and discover novel modifiers of CGGexp-related RAN translation. The RNA-tagging approach allowed us to reveal 172 proteins potentially binding to *FMR1* RNA including factors with translation regulatory properties. Some identified proteins overlapped with ones already described as binding to mutant *FMR1* mRNA such as SRPK1 or TAR DNA-binding protein 43 (TDP-43; *Malik et al., 2021b*; *Rosario et al., 2022*; *Sellier et al., 2010*; *Jin et al., 2007*; *Cid-Samper et al., 2018*; *Sellier et al., 2017*). Further, we described three new modifiers of RAN translation of this mRNA. Depletion of DHX15 helicase and two components of the 40 S subunit, RPS26 and RPS25, significantly decreased FMRpolyG levels (*Figure 1C* and *Figure 6A and D*). Importantly, silencing RPS26 alleviated the aggregation phenotype and slowed the apoptosis process caused by FMRpolyG expression (*Figure 1E&F*).

In yeast, the presence or absence of Rps26 in the ribosomal structure leads to the expression of different protein pools influencing temporal protein homeostasis in response to environmental stimuli (*Ferretti et al., 2017*; *Yang and Karbstein, 2022*). For instance, high-salt and high-pH stress induce the release of Rps26 from mature ribosomes by its chaperone Tsr2, enabling the translation of mRNAs engaged in stress response pathways (*Yang and Karbstein, 2022*). A previous study indicated that RAN translation in human cells is selectively enhanced by activating stress pathways in a feed-forward loop (*Green et al., 2017*). Therefore, differences in concentration of ribosomal components may influence FMRpolyG production under certain stress conditions.

In eukaryotic cells, RPS26 is involved in stress responses, however on the contrary to yeast model, RPS26 remains associated to the ribosome under energy stress (*Havkin-Solomon et al., 2023*). It was demonstrated that cells with mutated C-terminus of RPS26 were more resistant to glucose starvation, than the wild type cells (*Havkin-Solomon et al., 2023*). Moreover, RPS26 was shown to be involved in other cellular processes such as the DNA damage response (*Cui et al., 2014*), activation of the mTOR signaling pathway (*Havkin-Solomon et al., 2023*) and cellular lineage differentiation by preferential translation of certain transcripts (*Piantanida et al., 2022*; *Li et al., 2022*). We evaluated global changes in the human cellular proteome under RPS26 depletion and found that the expression level was significantly changed in approximately 20% of human proteins, while most proteins' expression levels remained intact (*Figure 3A*, *Supplementary file 1, table 7*). This suggests that many proteins, including FMRP, are not negatively affected by depleting RPS26, although FMRpolyG biosynthesis appears to be RPS26 level-sensitive. Our observation aligns well with current knowledge about how deficiency of different ribosomal proteins alters translation of some classes of mRNAs (*Luan et al., 2022*; *Cheng et al., 2019*). It was shown that depletion of RPS26 affects translation rate of different mRNAs compared to depletion of other proteins of small ribosomal subunit. We also verified whether RPS26 silencing would affect FMRP endogenously expressed in FXTAS patient-derived and control fibroblasts, as they differ in CGG repeats content in natural locus of *FMR1,* which potentially might influence the effect of RPS26 deficiency. However, we did not observe any changes in FMRP levels upon RPS26 silencing in either genetic variant with the PM or normal *FMR1* allele (*Figure 3F*).

Given the role of TSR2 in incorporating or replacing RPS26 into the ribosome, we investigated whether the depletion of TSR2 modulates FMRpolyG production. In line with previous studies (*Schütz et al., 2014*), we demonstrated that the RPS26 level decreases upon TSR2 silencing. We subsequently showed that TSR2 silencing incurs the same effect on RAN translation efficiency as the direct depletion of RPS26 and the observed regulation does not stem from global translation inhibition (*Figure 5A&C*). However, it remains unclear whether this is an effect of hampering RPS26 loading to the 40 S subunit by TSR2 depletion, or a consequence of decrease of RPS26 level. The silencing of this chaperone also exhibited selectivity towards FMRpolyG over the FMRP reading frame without affecting the other 40 S ribosomal proteins, such as RACK1, RPS6, and RPS15 (*Figure 5A&B*). Previously, it was shown that Tsr2 is responsible for replacing damaged Rps26, which undergoes oxidation on mature 80 S ribosomes under stress conditions (*Yang et al., 2023*). Therefore, the activity of this chaperone may play an important role in modulating RAN translation *via* Rps26 assembly or disassembly from ribosomal subunits, also during different stresses (*Yang and Karbstein, 2022*). Importantly, we provided additional evidence that RPS26 specifically affects CGGexp-related RAN translation, as the depletion of another 40 S ribosomal protein, RPS6, did not affect the FMRpolyG biosynthesis (*Figure 5C*). Moreover, mutations in genes encoding RPS26 and TSR2 were associated with hematopoiesis impairment that underlies the genetic blood disorder Diamond-Blackfan anemia (DBA) (*Piantanida et al., 2022*; *Li et al., 2023*; *Doherty et al., 2010*). Previously, analyses of DBA patient cells and RPS26-depleted HeLa cells found alterations in ribosome biogenesis and pre-rRNA processing (*Doherty et al., 2010*). Similarly, we found that proteins implicated in the eukaryotic PIC and components of large ribosomal subunits were altered upon RPS26 silencing in HEK293T cells (*Figure 4B*), suggesting rearrangements in ribosomal composition. Similar observation was noted in case of RPS25 knockdown which resulted in changes in abundance of ribosomal components (*Luan et al., 2022*). Hence, our data is consistent with the fact that mutants of 40 S components accumulates more 60 S subunits (*Cheng et al., 2019*) and the knockdown of 40 S ribosomal proteins elicits the enhanced translation of 60 S ribosomal proteins (*Luan et al., 2022*). Altogether, our data supports the evidence regarding co-dependent regulation of stability and accumulation of 40 S and 60 S ribosomal proteins (*Gregory et al., 2019*).

Moreover, RPS26 is highly expressed in ovarian cells and was shown to be important for female fertility (*Liu et al., 2018*). It is necessary for oocyte growth and follicle development, and its depletion causes changes in transcription and chromatin configuration, leading to premature ovarian failure (*Liu et al., 2018*). Our discovery that RPS26 regulates the level of FMRpolyG in cellular models sheds new light on the potential role of RPS26-related RAN translation in FXPOI, where FMRpolyG aggregates are detected in ovarian cells and contribute to the development and progression of fertility issues (*Buijsen et al., 2016*; *Shelly et al., 2021*; *Rosario et al., 2022*).

Given these findings, it is likely that the depletion of RPS26 from 40 S subunits contributes to its specialization and modulates the translation of specific mRNAs (*Yang and Karbstein, 2022*; *Ferretti et al., 2017*; *Li et al., 2022*; *Gaikwad et al., 2021*). Remarkably, our experiments identified nearly 400 proteins responding to RPS26 depletion, which is similar to ca. 500 mRNAs, for which translation rate was altered upon Rps26 insufficiency in yeast (*Gaikwad et al., 2021*). In fact, depletion or mutations in RPS26 resulted in the reduction of 40 S subunits level (*Gaikwad et al., 2021*; *Havkin-Solomon et al., 2023*); however, overall translation rate was not impacted by RPS26 impairment (*Havkin-Solomon et al., 2023*). We also observed that RPS26 depletion did not elicit global translation inhibition (*Figure 2B*), which additionally strengthens our conclusion that decrease of polyglycine production is not a byproduct of global translation shut down but rather the specific response to RPS26 deficiency. Alternatively, the observed effect may be explained by the decrease of translationally active ribosomes, which in return, impacts the translation rate of selected mRNAs, including upstream ORF of *FMR1*. In fact, it was shown that RPS26 insufficiency resulted in the decrease of mammalian cells viability (*Havkin-Solomon et al., 2023*), which is in line with our analysis regarding cells proliferation upon RPS26 silencing (*Figure 2A*). Moreover, RPS6 and RPS25 depletion also negatively impacted cells proliferation (*Figures 5D and 6B*), which is consistent with previously described data linking the ribosomal proteins deficiencies (both from 40 S and 60 S subunits) with a cellular growth rate inhibition (*Cheng et al., 2019*; *Luan et al., 2022*).

It has been shown that the C-terminal domain of RPS26 is essential for mRNA interaction (*Havkin-Solomon et al., 2023*), although whether RPS26 recognizes specific sequential or structural motifs within the mRNA remains unclear. Data derived from yeast models suggest that nucleotides in

positions −1 to −10 upstream of the AUG codon, especially Kozak sequence elements, play an important role in Rps26 interactions and translation efficiency (*Ferretti et al., 2017*). According to the established ribosome-mRNA structure, RPS26 contacts with mRNA upstream to AUG codon and the C-terminus reaches into mRNA exit channel (*Anger et al., 2013*; *Hussain et al., 2014*). Recent findings indicate that in eukaryotic cells, positions −11 to −16 upstream of the start codon might be more significant for RNA recognition, stabilizing PIC, and translation initiation (*Havkin-Solomon et al., 2023*). Our bioinformatic analysis of proteins sensitive to RPS26 depletion revealed that their transcripts were GC-rich (especially in the 5'UTR region) and enriched with *k*-mers mainly consisting of Gs and Cs (*Figure 4D*, *Figure 4—figure supplement 1*); however, we did not identify the importance of any specific sequence positions from AUG codon. These data and the fact that *FMR1* 5'UTR is a GC-rich sequence (~90%) suggest that RPS26 level-sensitive translation is selective for transcripts rich with GC nucleotides. Moreover, we demonstrated that RPS26 depletion affected RAN translation initiated from ACG but not CTG and GTG codons (*Figure 4F*), which indicates that the start codon selection step may constitute an important part of RPS26 level-sensitive RAN translation regulation. Such hypothesis can be supported by the fact that blockage of ACG codon with ASO caused less effective RAN translation downregulation evoked by RPS26 depletion (*Figure 4G*). This may point to the importance of specific RNA structural motifs localized within 5'UTRs or the speed of PIC scanning, which depends on how effectively stable secondary/tertiary structures are resolved. Other explanation would be differences in dynamics of either scanning of PICs differing in the presence of RPS26 or assembling of 80 S on the initiation codon.

The ribosomal protein RPS25, a component of 40 S subunit was previously described in the context of GGGGCC and CAG repeats-related RAN translation corresponding to C9-ALS/FTD, HD, and SCA (*Yamada et al., 2019*). The depletion of RPS25 in the *Drosophila* C9orf72 model, and in induced motor neurons derived from C9-ALS/FTD patients, alleviated toxicity caused by RAN translation (*Yamada et al., 2019*). On the contrary, in FXTAS *Drosophila* model, RPS25 knockdown led to enhancement of CGG repeats-related toxicity; however, underlying mechanism was not determined (*Linsalata et al., 2019*). Here, we demonstrated that RPS25 coprecipitated with the 5'UTR of *FMR1* and its depletion negatively affected the biosynthesis of FMRpolyG (*Figure 6A and D*; *Figure 6—figure supplement 1*), thus expanding current knowledge about RPS25 RAN translation modulatory properties in REDs.

Altogether, we have identified two ribosomal proteins, RPS26 and RPS25 as modifiers of CGGexp-related RAN translation of mutant *FMR1* mRNA, which imply that the downregulation of these two 40 S ribosomal proteins affects FMRpolyG biosynthesis. Importantly, we demonstrated that RPS26 depletion alleviated toxicity caused by FMRpolyG but did not affect FMRP, the main product of the *FMR1* gene. This suggests that sequence/structure elements within *FMR1* mRNA, which make this transcript sensitive to the level of studied ribosomal proteins, may be potential therapeutic targets.

## Materials and methods
### Genetic constructs

For MS-based screening *FMR1 RNA* construct was generated based on backbone described in *Sellier et al., 2017* (see also Addgene plasmid #63091), which contains 5'UTR of *FMR1* with expanded 99xCGG repeats. Enhanced green fluorescent protein (EGFP) sequence was removed and replaced with 4 STOP codons in FMRpolyG frame. Three times repeated MS2 stem loop aptamers (3xMS2 stem loops) were amplified by Phusion polymerase (Thermo Fisher Scientific) with primers introducing EagI restriction site from the plasmid (Addgene, #35572, *Tsai et al., 2011*). After gel-purification, PCR product was digested with EagI (New England Biolab) and inserted into EagI-digested and dephosphorylated backbone (CIAP; Thermo Fisher Scientific) downstream of 5'UTR of *FMR1* using T4 ligase (Thermo Fisher Scientific). To generate *GC-rich RNA* construct, GC rich sequence (corresponding to *TMEM170* mRNA, Supplementary file - sequences) was amplified by PCR introducing EagI restriction site (CloneAmp HiFi PCR Premix, TakaraBio). PCR product was digested by EagI (New England Biolabs) and ligated into *FMR1 RNA* construct backbone instead of 5'UTR of *FMR1* sequence using T4 ligase (Thermo Fisher Scientific). FMRpolyG-GFP (named here FMR99xG) transient expression was derived from 5'UTR CGG 99 x FMR1-EGFP vector (Adggene plasmid #63091), a kind gift from N. Charlet-Berguerand. In case of 5'UTR of *FMR1* with 16xCGG repeats tagged with nanoluciferase, all mutants were obtained using inverse PCR with primers containing 15 nucleotide long overhangs

(In-Fusion cloning, Takara Bio). PCR products were run on the agarose gel, appropriate bands were cut, and DNA was purified and used in a reaction with HiFi NEBuilder mix (New England Biolab) according to the manufacturer's instructions. Sequence of all constructs as well as content of CGG repeats were verified by Sanger sequencing.

## Generation of cell lines stably expressing FMRpolyG

To generate cell lines with doxycycline-inducible, stable FMRpolyG expression we used the Flp-In T-REx system. Through this approach, we obtained integration of constructs containing 5'UTR FMR1 with either 95 or 16 CGG repeats fused to *EGFP* into genome of 293 Flp-In T-Rex cells. These cell lines were named *S-95xCGG* and *S-16xCGG*, respectively, and expressed RAN proteins referred to as FMR95xG or FMR16xG. A detailed protocol concerning experimental procedure was described previously by *Szczesny et al., 2018*. The constructs used in the procedure were generated by modyfing pKK-RNAi-nucCHERRYmiR-TEV-EGFP (Addgene plasmid #105814). The insert containing CGG repeats within the 5'UTR of FMR1 gene fused with *EGFP* sequence was derived from 5'UTR CGG 99 x FMR1-EGFP (Addgene plasmid #63091). The insert was ligated into the vector instead of *EGFP* sequence. In the second approach, we used lentiviral transduction system using constructs containing 5'UTR of *FMR1* with 99 CGG repeats fused to EGFP followed by T2A autocleavage peptide and puromycin resistance (FMRpolyG-GFP_T2A_PURO). Cloning FMRpolyG-GFP_T2A_PURO sequence into TetO-FUW-Ascll-Puro vector (Addgene plasmid #97329, *Yang et al., 2017*) as well as lentiviral particles production was performed by The Viral Core Facility, part of the Charité – Universitätsmedizin Berlin. For transduction procedure $0.15x10^6$ HEK293T cells were plated in six-well plates at ~60% confluency and incubated with lentiviral particles for 48 hr. Subsequently, in order to obtain the pool of cells with integrated transgene, selection with puromycin (in final concentration 1 µg/ml, Sigma) was initiated and lasted for 72 hr, until all transgene-negative cells died. After selection we obtained polyclonal cell line named *L-99xCGG* expressing FMR99xG. Cell lines generated for the purpose of this study are available upon request.

## Cell culture and transfection

The monkey COS7, human HEK293T, HeLa and SH-SY5Y cells, purchased from ATCC, were grown in a high glucose DMEM medium with L-Glutamine (Thermo Fisher Scientific) supplemented with 10% fetal bovine serum (FBS; Thermo Fisher Scientific) and 1 x antibiotic/antimycotic solution (Sigma). *S-95xCGG* and *S-16xCGG* cells were grown in DMEM medium containing certified tetracycline-free FBS (Biowest). Fibroblasts derived from FXTAS male patient (1022–07) with $(CGG)_{81}$ in *FMR1* gene, and control, non-FXTAS male individual (C0603) with $(CGG)_{31}$ in *FMR1* were cultured in MEM medium (Biowest) supplemented with 10% FBS (Thermo Fisher Scientific), 1% MEM non-essential amino acids (Thermo Fisher Scientific) and 1 x antibiotic/antimycotic solution (Sigma). All cells were grown at 37 °C in a humidified incubator containing 5% $CO_2$. FXTAS 1022–07 (XY, 81xCGG, age 68 years old) line was a kind gift from P. Hagerman (*Garcia-Arocena et al., 2010*) while control C0603 fibroblast line (XY, 31xCGG) were given by A. Bhattacharyya (*Rovozzo et al., 2016*). Used cell lines were free of mycoplasma (verified by MycoStrip test [InvivoGen]). For the delivery of siRNA with the final concentration in culture medium ranging from 15 to 25 nM, reverse transfection protocol was applied using jetPRIME reagent (Polyplus) with the exception of fibroblasts, which were plated on the appropriate cell culture vessels the day before the transfection. Plasmids were delivered 24 hr post siRNA treatment, and the DNA/jetPRIME reagent (Polyplus) ratio 1:2 was applied. For the experiment from *Figure 4G*, 3 hr post plasmid encoding FMR99xG delivery, cells were transfected with ASOs with the final concentration of 200 nM, composed of LNA units. ASO_ACG sequence: 5'-CGTCACCGCCCG-3' and ASO_CTRL 5'-TTGAACATAAG-3'. Cells were harvested 48 hr post siRNA silencing and 24 hr post transient plasmids expression. The list of all siRNAs used in the study is available in a *Supplementary file 1, table 1*.

## Mass-spectrometry-based proteins screening

To capture RNA-protein complexes natively assembled within *FMR1 RNA* we adapted the MS2 in vivo biotin tagged RNA affinity purification (MS2-BioTRAP) technique, originally published by *Tsai et al., 2011*. The principle of this technique is to co-express bacteriophage MS2 protein fused to a HB tag which undergoes biotinylation in vivo, and RNAs tagged with so-called MS2 stem loop RNA aptamers, towards which MS2-HB protein represent high affinity. Natively assembled RNA-protein

complexes can be then fixed and identified with HB-tag based affinity purification using streptavidin-conjugated beads. To perform the screening, $2 \times 10^6$ HEK293T cells were co-transfected with genetic constructs: 2 µg of MS2-HB plasmid (#35573, *Tsai et al., 2011*) along with 8 µg of *FMR1 RNA* or *GC-rich RNA* encoding vectors (described in detail above). Three 10 cm plates were used per given mRNA with MS2 stem loop aptamers. 24 hr post co-transfections cells were washed two times with ice-cold phosphate buffered saline (PBS) and crosslinked with 0.1% formaldehyde (ChIP-grade, Pierce) for 10 min, followed by 0.5 M glycine quenching for 10 min in room temperature (RT). After one wash with ice-cold PBS cells were lysed for 30 min on ice in cell lysis buffer (50 mM Tris-Cl pH 7.5, 150 mM NaCl, 1% Triton X-100, 0.1% Na-deoxycholate) with Halt Protease Inhibitor Cocktail (Thermo Fisher Scientific) and RNAsin Plus (Promega) and vortexed. Next, cells lysates were sonicated (15 cycles: 10 s on/10 s off) using sonicator (Bioruptor, Diagenode) and centrifuged at $12,000 \times g$ for 10 min at 4 °C. Precleared protein extracts were transferred to Protein Lobind tubes (Eppendorf) and incubated with pre-washed 50 µl of magnetic-streptavidin beads (MyOne C1, Sigma) for 1.5 hr in cold room rotating. After immunoprecipitation (IP) procedure, beads were washed four-times for 5 min each on rotator: 1-time with 2% sodium dodecyl sulfate (SDS), 1-time with cell lysis buffer, 1-time with 500 mM NaCl, and last wash with 50 mM Tris-Cl pH 7.5. Finally, 20% of IP fraction was saved for western blot and remaining beads were submerged into digestion buffer (6 M Urea, 2 M Thiourea, 100 mM Tris, pH 7.8 and 2% amidosulfobetaine-14) and were shaken for 1 hr at RT. Then, samples were reduced using dithioerythritol for 1 hr at RT, and alkylated with iodoacetamide solution for 30 min in dark. Then, Trypsin/Lys-C (Promega) solution was added, and samples were incubated for 3 hr at 37 °C, followed by adding fresh Milli-Q water to dilute Urea to ~1 M and samples were further incubated at 37 °C overnight. Finally, beads were removed on a magnet, and peptides transferred to the clean tube, and desalted using C18 Isolute SPE columns (Biotage). Samples were analyzed in Mass Spectrometry Laboratory, Institute of Biochemistry and Biophysics, Polish Academy of Sciences, Pawińskiego 5 a Street, 02–106 Warsaw, Poland, using LC-MS system composed of Evosep One (Evosep Biosystems, Odense, Denmark) directly coupled to a Orbitrap Exploris 480 mass spectrometer (Thermo Fisher Scientific, USA). Peptides were loaded onto disposable Evotips C18 trap columns (Evosep Biosystems, Odense, Denmark) according to manufacturer protocol with some modifications. Briefly, Evotips were activated with 25 µl of Evosep solvent B (0.1% formic acid in acetonitrile, Thermo Fisher Scientific, USA), followed by 2 min incubation in 2-propanol (Thermo Fisher Scientific, USA) and equilibration with 25 µl of solvent A (0.1% FA in $H_2O$, Thermo Fisher Scientific, Waltham, Massachusetts, USA) Chromatography was carried out at a flow rate 220 nl/min using the 88 min gradient on EV1106 analytical column (Dr Maisch C18 AQ, 1.9 µm beads, 150 µm ID, 15 cm long, Evosep Biosystems, Odense, Denmark). Data was acquired in positive mode with a data-dependent method. MS1 resolution was set at 60,000 with a normalized AGC target 300%, Auto maximum inject time and a scan range of 350–1400 m/z. For MS2, resolution was set at 15,000 with a Standard normalized AGC target, Auto maximum inject time and top 40 precursors within an isolation window of 1.6 m/z considered for MS/MS analysis. Dynamic exclusion was set at 20 s with allowed mass tolerance of ±10 ppm and the precursor intensity threshold at 5e3. Precursor were fragmented in HCD mode with normalized collision energy of 30%.

## Biotinylated RNA-protein pull down

Biotinylated RNA probes were produced by in vitro transcription using T7 RNA polymerase (Promega), and by adding 1:10 CTP-biotin analog:CTP to the reaction mix, to incorporate biotinylated cytidines in random manner. Cellular extracts were prepared by lysing $2.5 \times 10^6$ HEK273T cells in 200 µL of mammalian cell lysis buffer. Lysates were cleared by centrifugation and 200 µL of supernatant was incubated for 20 min at 21 °C with 5 µg of RNA in 200 µL of 2xTENT buffer (100 mM Tris pH 7.8, 2 mM EDTA, 500 mM NaCl, 0.1% Tween) supplemented with RNAsin (Promega). RNA-protein complexes were then incubated with MyOne Streptavidin T1 DynaBeads (Thermo Fisher Scientific) for 20 min, followed by washing steps in 1 x TENT buffer. Bound proteins were released in Bolt LDS Sample Buffer (Thermo Fisher Scientific) followed by heat denaturation at 95 °C for 10 min. Samples were analyzed by SDS-PAGE and western blotting.

## Cell proliferation assay

Cells were seeded at the density of ca. $1 \times 10^5$ cells per well and treated with siRNA using reverse transfection protocol. Forty eight or 50 hr post silencing, cells were harvested and incubated with trypan

blue (Thermo Fisher Scientific). Subsequently, alive and dead cells were calculated using Countess 3 (Thermo Fisher Scientific) equipment.

## Cell viability assay

Cells were seeded at the density of $1\times10^5$ cells per well and treated with siRNA using reverse transfection protocol. Subsequently, reagents from RealTime-Glo MT Cell Viability Assay (Promega) were added to the culture. Measurement of luminescence signal corresponding to cell viability was taken upon 24 and 48 hr post siRNA treatment using SPARK microplate reader (TECAN).

## Apoptosis assay

For luminescent-based apoptosis assay $1\times10^4$ HeLa cells were seeded on 96-well plate and transfected with siRPS26 and siCtrl in final concentration of 15 nmol. In order to induce FMRpolyG derived toxicity, after 24 hr post silencing, cells well transfected with plasmid encoding FMR99xG or with jetPRIME reagent (Polyplus) as a MOCK control. Subsequently, reagents from RealTime-Glo Annexin V Apoptosis Assay (Promega) were added to the culture. Measurement of luminescence signal corresponding to apoptosis progression was taken upon 28 and 43 hr post FMR99xG expression using SPARK microplate reader (TECAN).

## Nanoluciferase assay

For luminescence-based analysis of RAN translation post RPS26 silencing, $1\times10^4$ of HEK293T cells were seeded on 94-well plate and transfected with siRPS26 using reverse transfection protocol. On the next day, cells were cotransfected with constructs expressing FMRpolyG (16xCGG) in frame with nanoluciferase together with the construct expressing firefly luciferase used for normalization purposes. 24 hr post plasmid delivery, cells were lysed for 1 hr on ice in RIPA buffer (Merck). Subsequently, 1/5 of the lysate was transferred to black 96 well plate (Thermo Fisher Scientific) and incubated with ONE-Glo EX reagent (Promega). After 10 min incubation, firefly luciferase-based luminescence was measured by SPARK microplate reader (TECAN). Next, the NanoDLR Stop & Glo (Promega) buffer containing substrate for nanoluciferase was added and after 10 min incubation, the nanoluciferase-based luminescence was measured by the same equipment.

## Microscopic analysis

To detect aggregated form of FMRpolyG fluorescence microscopy experiments were performed as described previously (*Derbis et al., 2018*). Briefly, before analysis, HeLa cells were grown on 48-well plates in standard growth medium, siRNA and plasmid transfections were performed as described in Cell culture and transfection section. Prior microscopic analysis, final concentration of 5 µg/ml of Hoechst 33342 (Thermo Fisher Scientific) was added to cell culture and incubated for 10 min. Images were taken with Axio Observer.Z1 inverted microscope equipped with A-Plan 10×/0.25 Ph1 or LD Plan-Neofluar 20×/0.4 Ph2 objective (Zeiss), Zeiss Colibri 7 excitation band 385/30 nm, emission filter 425/30 nm (Hoechst) and Zeiss Colibri 7 excitation band 469/38 nm, emission filter 514/30 nm (GFP), Zeiss AxioCam 506 camera and ZEN 2.6 pro software, 48 hr post siRNA transfection. Presented values were quantified from 10 microscopic images for each condition, number of cells (blue signal) and aggregates (dense green signal) were calculated using ImageJ and AggreCount plugin (*Klickstein et al., 2020*), which localizes cellular aggregates in relation to the nuclei. Each image contained ca. 1100 cells and the number of aggregates per nucleus were calculated as follows: nuclei were processed via fluorescent signal enhancement, converted to binary image and region of interest (ROI) were captured, size filter was set to $100-6\times10^6$ voxels; aggregates were detected used difference of gaussians methodology, size filter was set to 10–500 voxels. Cells were segmented using the watershed-based segmentation algorithm. For each image, the number of detected aggregates was divided per number of nuclei, and the results for all images were plotted on a histogram. To validate FMR95xG or FMR16xG expression in *S-95xCGG* and *S-16xCGG* models, microscopic analysis was performed at two time points: 4 days and 35 days post doxycycline (DOX) induction. 4 days post doxycycline induction cells were incubated in a cell culture medium with Hoechst 33342 (Thermo Fisher Scientific) at a final concentration 5 µg/ml for 30 min at 37 °C. Images were taken as described above. 35 days post doxycycline induction images were captured with Leica Stellaris 8 Inverted Confocal Microscope equipped with HC PL APO CS2 63 x/1.20 water objective and an onstage incubation

chamber controlling temperature and $CO_2$ concentration. GFP was excited with 489 nm laser and detected with Power HyD S detector (spectral positions: 494 nm - 584 nm). To visualize FMRpolyG positive aggregates the *L-99xCGG* model cells were incubated with Hoechst 33342 (Thermo Fisher Scientific) at a final concentration 5 µg/ml for 15 min at 37 °C. Images were taken with Leica SP8 confocal microscope equipped with White Light Laser 2 PP, Diode Laser 405 nm, HyD S detectors, and high precision objective (PL APO 63 x/1.20 W CORR CS2). GFP was excited with 489 nm and detected in range of 500–550 nm. Hoechst was excited with 405 nm laser lines and detected in 420–470 nm detector range.

## RNA isolation and quantitative real-time RT-PCR

Cells were harvested in TRI Reagent (Thermo Fisher Scientific) and total RNA was isolated with Total RNA Zol-Out D (A&A Biotechnology) kit according to the manufacturer's protocol. 500–1000 ng of RNA was reversely transcribed using GoScript Reverse Transcriptase (Promega) and random hexamers (Promega). Quantitative real-time RT-PCRs were performed in a QuantStudio 7 Flex System (Thermo Fisher Scientific) using Maxima SYBR Green/ROX qPCR Master Mix (Thermo Fisher Scientific) with 5 ng of cDNA in each reaction. Transgene *FMR1-GFP* mRNAs with expanded CGG repeats were amplified with primers: Forward: 5′ GCAGCCCACCTCTCGGGG 3′, Reverse: 5′ CTTCGGGCATGGCGGACTTG 3′ with a note that reverse primer was anchored in GFP sequence in order to distinguish endogenously expressed *FMR1* transcripts (amplified with primer pair: Forward: 5′ TGTGTCCCCATTGTAAGCAA 3′, Reverse: 5′ CTCAACGGGAGATAAGCAG 3′). Reactions were run at 58 °C annealing temperature and Ct values were normalized to *GAPDH* mRNA level (amplified with primer pair: Forward: 5′ GAGT CAACGGATTTGGTCGT 3′, Reverse: 5′ TTGATTTTGGAGGGATCTCG 3′). Fold differences in expression level were calculated according to the $2^{-\Delta\Delta Ct}$ method (*Livak and Schmittgen, 2001*).

## SDS-PAGE and western blot

Cells were lysed in cell lysis buffer supplemented with Halt Protease Inhibitor Cocktail (Thermo Fisher Scientific) for 30 min on ice, vortexed, sonicated and centrifuged at 12,000 ×*g* for 10 min at 4 °C. Protein lysates were heat-denatured for 10 min at 95 °C with the addition of Bolt LDS buffer (Thermo Fisher Scientific) and Bolt Reducing agent (Thermo Fisher Scientific) and separated in Bolt 4–12% Bis-Tris Plus gels (Thermo Fisher Scientific) in Bolt MES SDS Running Buffer (Thermo Fisher Scientific). Next, proteins were electroblotted to PVDF membrane (0.2 µM, GE Healthcare) for 1 hr at 100 V in ice-cold Bolt Transfer Buffer (Thermo Fisher Scientific). Membranes were blocked in room temperature for 1 hr in 5% skim milk (Sigma) in TBST buffer (Tris-buffered saline [TBS], 0.1% Tween 20) and subsequently incubated overnight in cold room with primary antibody solutions diluted in blocking buffer (the list of all commercially available primary antibodies used in the study including catalog numbers is presented in a *Supplementary file 1, table 2*, with the exception for home-made anti-FUS antibody *Raczynska et al., 2015*). The following day membranes were washed 3-times with TBST for 7 min and incubated with corresponding solutions of anti-mouse (A9044, Sigma; 1:15,000) or anti-rabbit (A9169, Sigma; 1:20,000) antibodies conjugated with horse radish peroxidase (HRP) for 1 hr in RT. For Vinculin and GFP detection, membranes were incubated with antibodies already conjugated with HRP (*Supplementary file 1, table 2*) overnight in cold room. After final washing steps signals were developed using Immobilon Forte Western HRP substrate (Sigma) using G:Box Chemi-XR5 (Syngene) and ChemiDoc Imaging System (Bio-Rad) and quantified using Multi Gauge 3.0 software (Fujifilm). Briefly, area of bands corresponding to proteins of interest and to background were manually selected for measurement. Then, the signal of background was subtracted and the percentage (gray-scale density) of each band was calculated. Relative protein level was normalized to Vinculin.

## Surface sensing of translation (SuNseT) assay

48 hr post siRNA transfection HEK293T cells were incubated with puromycin (PURO; Sigma) at a final concentration of 10 µg/ml, which allowed to incorporate PURO into newly synthesized peptides to asses global translation rate (*Schmidt et al., 2009*). After 10 min incubation with PURO cells were subsequently harvested. For positive control of translation inhibition cells were incubated for 5 min with cycloheximide (CHX; 100 µg/ml) prior addition of puromycin. DMSO treated cells without puromycin incubation served as a negative control. Protein lysates were prepared as described above. Next, protein concentration was determined by BCA assay (Pierce) and 10 µg of protein lysate was

loaded on the Bolt 4–12% Bis-Tris Plus gel (Thermo Fisher Scientific). Following steps of western blot were described as above. Anti puromycin antibody (1:5000 (clone 12D10, Merck), overnight incubation at 4 °C) was used to detect puromycin-incorporated peptides.

## Stable isotope labeling using amino acids in cell culture (SILAC) coupled with MS

To quantify changes in protein levels upon RPS26 silencing we applied SILAC in HEK293T cells grown in light and heavy amino acids ($^{13}C_6$ $^{15}N_2$ L-lysine-2HCl and $^{13}C_6$ $^{15}N_4$ L-arginine-HCl) containing media (Thermo Fisher Scientific). Cells were cultured in light or heavy media for more than 14 days to ensure maximum incorporation of labeled amino acids. Subsequently, cells were transfected with siRPS26 or siCtrl in final concentration of 15 nM and harvested 48 hr post silencing. Sample preparation was performed using modified filter-aided sample preparation method as described previously (*Laakkonen et al., 2017*; *Wiśniewski et al., 2009*). Briefly, 10 µg of proteins were washed 8-times with 8 M Urea, 100 mM ammonium bicarbonate in Amicon Ultra-0.5 centrifugal filters, and Lysine-C endopeptidase solution (Wako) in a ratio of 1:50 w/w was added to the protein lysates followed by incubation at room temperature overnight with shaking. The peptide digests were collected by centrifugation and trypsin solution was added in a ratio of 1:50 w/w in 50 mM ammonium bicarbonate and incubated overnight at room temperature. The peptide samples were cleaned using Pierce C18 reverse-phase tips (Thermo Fisher Scientific). Dried peptide pellets were re-suspended in 0.3% Trifluoroacetic acid and analyzed using nano-LC-MS/MS in Meilahti Clinical Proteomics Core Facility, University of Helsinki, Helsinki, Finland. Peptides were separated by Ultimate 3000 LC system (Dionex, Thermo Fisher Scientific) equipped with a reverse-phase trapping column RP-2TM C18 trap column (0.075x10 mm, Phenomenex, USA), followed by analytical separation on a bioZen C18 nano column (0.075×250 mm, 2.6 µm particles; Phenomenex, USA). The injected samples were trapped at a flow rate of 5 µl/min in 100% of solution A (0.1% formic acid). After trapping, peptides were separated with a linear gradient of 125 min. LC-MS acquisition data was performed on Thermo Q Exactive HF mass spectrometer with following settings: resolution 120,000 for MS scans, and 15,000 for the MS/MS scans. Full MS was acquired from 350 to 1400 m/z, and the 15 most abundant precursor ions were selected for fragmentation with 45 s dynamic exclusion time. Maximum IT were set as 50 and 25ms and AGC targets were set to 3 e6 and 1 e5 counts for MS and MS/MS, respectively. Secondary ions were isolated with a window of 1 m/z unit. The NCE collision energy stepped was set to 28 kJ mol–1.

## MS data analysis

Raw data obtained from the LC-MS/MS runs were analyzed in MaxQuant v2.0.3.0 (*Cox and Mann, 2008*) using either the label-free quantification (LFQ, for MS2 pull down samples) or stable isotope labeling-based quantification (for SILAC-MS samples) with default parameters. UniProtKB database for reviewed human canonical and isoform proteins of May 2023 was used. The false discovery rate (FDR) at the peptide spectrum matches and protein level was set to 0.01; variable peptide modifications: oxidation (M) and acetyl (N-term), fixed modification: carbamidomethyl (C), label Arg10, Lys8 (for SILAC-MS samples only), two missed cleavages were allowed. Statistical analyses were performed using Perseus software v2.0.3.0 (*Tyanova et al., 2016*) after filtering for 'reverse', 'contaminant' and 'only identified by site' proteins. The LFQ intensity was logarithmized (log2[x]), minimum valid values were set to 70% in order to remove proteins which were not quantified, and imputation of missing values was performed with a normal distribution (width = 0.3; shift = 1.8). Proteomes were compared using t-test statistics with a permutation-based FDR of 5% and p-values <0.05 were considered to be statistically significant. The data sets, the Perseus result files used for analysis, and the annotated MS/MS spectra were deposited at the ProteomeXchange Consortium (*Deutsch et al., 2023*) *via* the PRIDE partner repository (*Perez-Riverol et al., 2022*) with the dataset identifier PXD047400 - MS2 pull down data and PXD047397 – SILAC-MS data.

## Gene ontology analysis

Gene ontology (GO) analysis performed on proteins that bind to *FMR1 RNA* (common between three replicates) was performed using an online tool - g:Profiler (*Raudvere et al., 2019*) with g:SCS algorithm, where at least 95% of matches above threshold are statistically significant. As reference proteome we used total human proteome. GO analysis performed on SILAC-MS data was performed

with PANTHER 18.0 (*Thomas et al., 2022*). Statistical significance was calculated using Fisher's Exact test with Bonferroni correction and only GO terms with p-values <0.05 were plotted. As reference proteome we used proteins named as non-responders (described in Results section).

## Bioinformatic analyses

The sequence logo analysis represents the frequency of nucleotides across transcript sequence positions in the close vicinity of start codon within three groups of transcripts: positive responders, negative responders, and background (BG, total transcriptome). The analyzed sequence positions span the last 20 nucleotides of the 5'UTR sequence (from −20 to −1 downstream of the start codon) and the first 20 nucleotides of the coding sequence (from +1 to+20 upstream of the start codon), forming 40-nucleotide sequence fragments that were used to generate the sequence logos with WebLogo v2.8.2 (*Crooks et al., 2004*). To calculate the GC content of 5'UTRs and CDSs, gene annotations for all protein-coding human genes (GRCh38.p14), including coding and 5'UTR sequences of transcripts, were obtained from Ensembl release 110 (Nov2023; *Martin et al., 2023*). UniProt protein accessions for positive and negative responders were mapped to corresponding Ensembl genes and transcripts using BioMart (*Smedley et al., 2009*). The reference BG dataset comprised all protein-coding genes, excluding positive and negative responder genes, with one randomly selected transcript per gene. p-values for pairwise comparisons of GC content between transcript groups were calculated using a two-tailed paired t-test with Bonferroni correction. For 5'UTRs lengths, a balanced comparison was ensured by randomly selecting 1000 transcripts from the complete human transcriptome to represent background. p-values for comparing 5'UTR sequence lengths between positive and negative responders relative to the background transcriptome were calculated using a one-tailed Mann-Whitney U test. Hexamer frequencies in 5'UTR sequences of the three datasets (i.e. positive and negative responders, and BG) were calculated using Jellyfish v2.3.1 (*Marçais and Kingsford, 2011*). The p-value associated with each hexamer in the positive and negative responder datasets, representing its overrepresentation compared to BG, was calculated from the binomial distribution. All statistical analyzes related to the comparison of nucleotide composition among the BG, positive and negative responder datasets were performed using SciPy v1.10.1 (*Virtanen et al., 2020*).

## Statistics

Group data are expressed as the means ± standard deviation (SD). Error bars represent SD. The statistical significance (if not indicated otherwise) was determined by an unpaired, two-tailed Student's t-test using Prism software v.8 (GraphPad): ∗, p<0.05; ∗∗, p<0.01; ∗∗∗, p<0.001; ∗∗∗∗, p<0.0001; ns, non-significant. All experiments presented in this work were repeated at least two times with similar results with at least three independent biological replicates (N=3).

## Acknowledgements

We thank A Bhattacharyya and P Hagerman for FXTAS and control fibroblasts. The 5'UTR CGG 99 x FMR1-EGFP construct was a gift from Nicolas Charlet-Berguerand (Addgene plasmid # 63091). We also thank Dominik Cysewski for participating in the preparation and analysis of MS samples from the MS2-based protein screening, Marc H Baumann and Rabah Soliymani from Meilahti Clinical Proteomics Core Facility for SILAC-MS samples analysis, Dorota Raczyńska for the kind gift of antibodies (anti-FUS and anti-RPS6), Wojciech Kwiatkowski for the assistance with the microscopic analyses, and Roman Szczęsny for the pKK-RNAi-nucCHERRYmiR-TEV-EGFP genetic construct. This work was supported by the Polish National Science Center [2019/35/D/NZ2/02158 to AB, 2020/38 /A/NZ3/00498 to KS] and the European Union's Horizon 2020 Research and Innovation Program under the Marie Sklodowska-Curie grant agreement [No. 101003385 to AB] and Initiative of Excellence–Research University at Adam Mickiewicz University, Poznan, Poland grant number [140/04/POB2/0006 to AB] KT held the Adam Mickiewicz University Foundation scholarship, awarded for the academic year 2023/24.

## Additional information

### Funding

| Funder | Grant reference number | Author |
|---|---|---|
| Polish National Science Center | 2019/35/D/NZ2/02158 | Anna Baud |
| Polish National Science Center | 2020/38/A/NZ3/00498 | Krzysztof Sobczak |
| Horizon 2020 Framework Programme | 101003385 | Anna Baud |
| Initiative of Excellence–Research University at Adam Mickiewicz University, Poznan | 140/04/POB2/0006 | Anna Baud |
| Adam Mickiewicz University Foundation | Scholarship for the best PhD Students | Katarzyna Tutak |

The funders had no role in study design, data collection and interpretation, or the decision to submit the work for publication.

### Author contributions

Katarzyna Tutak, Conceptualization, Investigation, Visualization, Writing – original draft, Writing – review and editing; Izabela Broniarek, Daria Niewiadomska, Investigation; Andrzej Zielezinski, Software, Investigation; Tomasz Skrzypczak, Visualization; Anna Baud, Conceptualization, Supervision, Funding acquisition, Investigation, Writing – original draft, Writing – review and editing; Krzysztof Sobczak, Conceptualization, Supervision, Funding acquisition, Writing – original draft, Writing – review and editing

### Author ORCIDs

Katarzyna Tutak http://orcid.org/0000-0002-2801-9956
Anna Baud http://orcid.org/0000-0003-3710-5722
Krzysztof Sobczak https://orcid.org/0000-0001-8352-9812

Reviewer #2 (Public review): https://doi.org/10.7554/eLife.98631.3.sa1
Reviewer #3 (Public review): https://doi.org/10.7554/eLife.98631.3.sa2
Author response https://doi.org/10.7554/eLife.98631.3.sa3

## Additional files

### Supplementary files

Supplementary file 1. **Table S1** (.xlsx file) List of siRNA used in the study. **Table S2** (.xlsx file) List of primary antibodies used in the study. **Table S3** (.xlsx file) Unfiltered list of proteins identified in MS2-based screening, as binding to FMR1-RNA and GC-rich RNA baits. **Table S4** (.xlsx file) Results of Gene ontology analysis performed on proteins bound to FMR1 RNA . **Table S5** (.xlsx file) Statistical analysis of filtered Label Free Quantification of MS2-based data (FMR1 RNA vs GC-rich RNA). Related to *Figure 1B*. **Table S6** (.xlsx file) Unfiltered list of proteins identified in SILAC-MS analysis of cells with knocked down RPS26 level (siRPS26, heavy-labelled) and cells treated with control siRNA (siCtrl, light-label). Normalized ratio of heavy/light signals and intensities for detected proteins are given. **Table S7** (.xlsx file) Statistical analysis of filtered SILAC-MS data. Related to *Figure 4A*. **Table S8** (.xlsx file) Results of Gene ontology analysis performed on selected group of proteins (negative and positive responders) identified in SILAC-MS data. **Table S9** (.xlsx file) Results of WebLogo analysis; frequencies of individual nucleotide at positions in the 20-nucleotide upstream or downstream to the start codon determined for transcripts groups (negative, positive responders and background). **Table S10** (.xlsx file) Lists of k-mers (hexamers) identified in 5'UTRs of negative and positive responders.

MDAR checklist

## Data availability

The mass spectrometry data generated and used in this article are available in ProteomeXchange Consortium via the PRIDE partner repository and can be accessed with the dataset identifier: PXD047400 - MS2 pull down data, PXD047397 - SILAC-MS data Supplementary data is deposited on Zenodo server under the https://doi.org/10.5281/zenodo.14954843. Code used to perform bioinformatic analyses is deposited on Gitlab platform under the link https://gitlab.com/source-code145207/RPS26/ (copy archived at *Zielezinski, 2025*).

The following datasets were generated:

| Author(s) | Year | Dataset title | Dataset URL | Database and Identifier |
|---|---|---|---|---|
| Webel H, Perez-Riverol Y, Nielsen AB, Rasmussen S | 2024 | The mass spectrometry data generated and used in this article are available in ProteomeXchange Consortium via the PRIDE partner repository | https://proteomecentral.proteomexchange.org/?search=PXD047400 | ProteomeXchange, PXD047400 |
| Webel H, Perez-Riverol Y, Nielsen AB, Rasmussen S | 2024 | The mass spectrometry data generated and used in this article are available in ProteomeXchange Consortium via the PRIDE partner repository | https://proteomecentral.proteomexchange.org/?search=PXD047397 | ProteomeXchange, PXD047397 |
| Tutak K, Sobczak K, Baud A | 2025 | Data underlying the article: Insufficiency of 40S ribosomal proteins, RPS26 and RPS25 negatively affects biosynthesis of polyglycine-containing proteins in fragile-X associated conditions | https://doi.org/10.5281/zenodo.14954843 | Zenodo, 10.5281/zenodo.14954843 |

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
