## [Editor Report · eLife Assessment]

In this **valuable** study, Tutak and colleagues set out to identify factors that mediate Repeat Associated Non-AUG (RAN) translation of CGG repeats in the FMR1 mRNA which are implicated in toxic protein accumulation that underpins ensuing neurological pathologies. The authors provide **solid** evidence that RPS26 may be implicated in mediating the RAN translation of FMR1 mRNA. This article should be of broad interest to researchers in the variety of disciplines including post-transcriptional regulation of gene expression and neurobiology.

---

## [Referee Report · Reviewer #2 (Public review)]

Summary:

Translation of CGG repeats leads to accumulation of poly G, which is associated with neurological disorders. This is an important paper in which the authors sought out proteins that modulate RAN translation. They determined which proteins in Hela cells were enriched on CGG repeats and affected levels of polyG encoded in the 5'UTR of the FMR1 mRNA. They then showed that siRNA depletion of ribosomal protein RPS26 results in less production of FMR1polyG than in control. Experiments were performed in several cell lines and with several reporters with differences in repeats and transfection methods to increase confidence that changes were occurring. New data and details of the methods increase confidence that reporter translation but not global translation is diminished by RPS26 knockdown as concluded. The manuscript has been improved by data showing that new proteins are being synthesized in cells following RPS26 knockdown, and that near-cognate start codon usage is diminished in lines when RPS26 is knocked down, but the mechanism by which RPS26 depletion affects translation is still unclear.

Strengths:

- The authors have proteomics data that show enrichment of a set of proteins on FMR1-polyG RNA but not a related RNA.

- Knockdown of RPS26, which was enriched on the FMR1 RNA, led to decreases in cell growth, but surprisingly did not strongly affect global translation, as assessed by puromycin incorporation

- There is some new evidence that near-cognate start codon selection is affected by RPS26 knockdown

Weaknesses:

- The mechanism for RPS26 knockdown affecting translation of the polyG sequences is unclear, whether knockdown is affecting ribosome levels, extra ribosomal RPS26 or ribosome composition is not known.

---

## [Referee Report · Reviewer #3 (Public review)]

Tutak et al provide intriguing findings demonstrating that insufficiency of RPS26 and related proteins, such as TSR2 and RPS25, downregulates RAN translation from CGG repeat RNA in fragile X-associated conditions. Using RNA-tagging system and mass spectrometry-based screening, the authors identified RPS26 as a potential regulator of RAN translation. They further confirmed its regulatory effects on RAN translation by siRNA-based knockdown experiments in multiple cellular disease models. Quantitative mass spectrometry analysis revealed that the expression of some ribosomal proteins is sensitive to RPS26 depletion, while approximately 80% of proteins, including FMRP, were not influenced. Given the limited understanding of the roles of ribosomal proteins in RAN translation regulation, this study provides novel insights into this research field. However, certain data do not fully support the authors' critical conclusions.

(1) While the authors substituted the ACG near-cognate initiation codon with other near-cognate codons, such as GTG and CTG, in the luciferase assay (Figure 4F), substitution of the ACG codon with an ATG codon should also be performed. Although they evaluated RPS26 knockdown effect on AUG-dependent FMRP translation in Figure 3C, investigating its effect on AUG-dependent repeat-associated translation (e.g., AUG-CGG-repeat) is necessary to substantiate their claim that ACG codon selection is important for RAN translation downregulation by RPS26 knockdown.

(2) The results of the ASO-based ACG codon-blocking experiment in Figure 4G are difficult to interpret. While RPS knockdown reduces FMRpolyG expression, the effect appears attenuated by the ASO-ACG treatment compared to the control. However, this does not conclusively demonstrate that the regulatory effect is directly due to ACG codon selection during translation initiation for some reasons. For example, ASO-ACG treatment possibly interferes with ribosomal scanning rather than ACG-codon selection, or alters the expression of template CGG repeat RNA. To validate the effect of RPS26 knockdown on ACG codon selection, experiments using the ACG-to-ATG substituted CGG repeat reporter are recommended, as suggested in comment 1.

(3) The regulatory effects of RPS26 and other molecules on RAN translation have been investigated as effects on the expression levels of FMRpolyG proteins upon knockdown of these molecules in disease model cells expressing CGG repeat sequences (Figures 1C, 1D, 3B, 3C, 3E, 4F, 4G, 5A, 5C, 6A, 6D). However, FMRpolyG expression levels can be influenced by factors other than RAN translation in these cellular experiments, such as template RNA level, template RNA localization, and FMRpolyG protein degradation. Although the authors evaluated the effect on the expression levels of template CGG repeat RNA, it would be better to confirm the direct effect of these regulators on RAN translation by other experiments. In vitro translation assay that can directly evaluate RAN translation is preferable, but experiments using the ACG-to-ATG substituted CGG repeat reporter, as suggested in comment 1, would also provide valuable insights.

---

## [Author Response]

The following is the authors’ response to the current reviews.

We thank Reviewers for highlighting the strengths of our work along with suggestions for future directions.

We agree with the Reviewers that RPS26 depletion may impact not only RAN translation initiation and codon selection (as showed in the experiments in Figure 4G), but also other mechanisms, such as speed of PIC scanning, as we stated in the discussion. Although, we did provide the data showing that mRNA of exogenous *FMR1-GFP* does not change upon RPS26 depletion (Figure 3B&C), hence observed effect most likely stems from translation regulation. In addition, an experiment with ASO-ACG treatment (Figure 4G) suggests that near cognate start codon selection or speed of PIC scanning may be a part of the regulation of RAN translation sensitive to RPS26 depletion. In addition, our latest unpublished results (Niewiadomska D. et al., in revision), indicate that FMRpolyG in fusion with GFP is fairly stable, in particular, while derived from long repeats (>90xCGG), suggesting that the protein stability is not at play in RPS26-dependent regulation.

We would like to stress that in order to avoid bias in result interpretation and to mimic the natural situation, the majority of experiments concerning levels of FMRpolyG were performed in cell models with stable expression of ACG-initiated FMRpolyG. Currently, we do not possess a cell model with stable expression of AUG-initiated FMRpolyG, and the experiments based on transient transfection system would not necessarily be comparable to the results obtained in stable expression system. However, we believe that the experiment presented in Figure 2B serves as a good control for overall translation level upon RPS26 depletion indicating that RPS26 insufficiency does not affect global translation and the observed regulation is specific to some mRNAs including the one encoding FMRpolyG frame. We also show that the level of ca. 80% of identified canonical proteins, including FMRP, did not change upon RPS26 silencing (SILAC-MS, Figure 4A). Indeed, we did not explore the ribosome composition upon RPS26 and TSR2 depletion, although, most likely the pool of functional ribosomes in the cell is sufficient enough to support the basal translation level (SUnSET assays, Figure 2B & 5C). However, we cannot exclude possibility that for some mRNAs, including one encoding for FMRpolyG, the observed effect can be partially caused by lowering the number of fully active ribosomes, especially in experiments with transient transfection experiments where transgene expression is hundreds times higher than for average native mRNA.

Finally, we agree with the Reviewer that in vitro translation assay would provide the evidence of direct effect of RPS26 on FMRpolyG level, however, we did not manage to overcome technical difficulties in obtaining cellular lysate devoid of RPS26 from vendor companies.

The following is the authors’ response to the original reviews.

General Comments

We thank Reviewers for the critical comments and experimental suggestions. We considered most of the advices in the revised version of the manuscript, which allowed for a more balanced interpretation of the results presented, and further supported major statement of the manuscript that insufficiency of the RPS26 and RPS25 plays a role in modulating the efficiency of noncanonical RAN translation from *FMR1* mRNA, which results in the production of toxic polyglycine protein (FMRpolyG). Firstly, performing new experiments, we showed that silencing of the RPS26 and its chaperone protein TSR2, which regulates loading/exchange of RPS26 in maturing small ribosome subunit, did not elicit global translation inhibition. Secondly, we demonstrated that in contrary to RPS26 and RPS25 depletion, silencing the RPS6 protein, a core component of 40S subunit, did not affect FMRpolyG production, further supporting the specific effect of RPS26 and RPS25 on RAN translation regulation of mutant FMR1 mRNA. We also observed that depletion of RPS26, RPS25 and RPS6 had significant negative effect on cells proliferation which is in line with previously published results indicating that insufficiencies of ribosomal proteins negatively affect cell growth. Moreover, we showed that FMRpolyG production is significantly affected by RPS26 depletion while initiated at ACG, but not other near cognate start codons. Importantly, translation of FMRP initiated at canonical AUG codon of the same mRNA upstream the CGGexp was not affected by RPS26 silencing, similarly to vast majority of the human proteome. This implies that RAN translation of *FMR1* mRNA mediated by RPS26 insufficiency is likely to be dependent on start codon selection/fidelity. In essence, we provide a series of evidences indicating that cellular amount of 40S ribosomal proteins RPS26 and RPS25 is important factor of CGGrelated RAN translation regulation. Finally, we also decided to tone down our claims. Now, we state that the RPS26/25/TSR2 insufficiency or depletion, affects RAN translation, rather than composition of 40S ribosomal subunit *per se* influences RAN translation. We have addressed all specific concerns below and made changes to the new version of manuscript.

**Public Reviews:**

**Reviewer #1 (Public Review):**
Summary:In this manuscript, Tutak et al use a combination of pulldowns, analyzed by mass spectrometry, reporter assays, and fluorescence experiments to decipher the mechanism of protein translation in fragile X-related diseases. The topic is interesting and important.Although a role for Rps26-deficient ribosomes in toxic protein translation is plausible based on already available data, the authors' data are not carefully controlled and thus do not support the conclusions of the paper.

We sincerely appreciate your rigorous, insightful, and constructive feedback throughout the revision process. We believe your guidance has been instrumental in significantly enhancing the quality of our research. Below, we have addressed your comments pointby-point.

Strengths:The topic is interesting and important.Weaknesses:In particular, there is very little data to support the notion that Rps26-deficient ribosomes are even produced under the circumstances. And no data that indicate that they are involved in the RAN translation. Essential controls (for ribosome numbers) are lacking, no information is presented on the viability of the cells (Rps26 is an essential protein), and the differences in protein levels could well arise from block in protein synthesis, and cell division coupled to differential stability of the proteins.

We agree that data presented in the first version of the manuscript did not directly address the following processes: ribosome content, global translation rate and cell viability upon RPS26 depletion. Therefore we addressed some of the issues in the revised version of the manuscript. In particular, we showed that RPS26 and TSR2 knock down did not inhibit global translation (new Figure 2B & 4C), hence we concluded that the changes of FMRpolyG level did not arise from general translational shut down. On the other hand, RPS26, RPS25 and RPS6 depletion negatively affected cells proliferation (new Figure 2A,5D,6C), which is in line with a number of previously published researches (e.g. Cheng *et al*, 2019; Havkin-Solomon *et al*, 2023). However, the rate of proliferation abnormalities is limited. We agree that observed effects on RAN translation from mutant FMR1 mRNA may stem from the combination of altered protein synthesis, conditions of the cells but also cis-acting factors of mRNA sequence/structure. In new experiments we showed that single nucleotide substitution of ACG by other near cognate start codons change sensitivity of RAN translation to insufficiency of RPS26 (new Figure 4F). Also the inhibitory effect of antisense oligonucleotide binding to the region of 5’UTR containing ACG initiation codon (ASO_ACG) is different in cells differing in amount of RPS26 (new Figure 4G).

We also agree that our data only partially supports the role of RPS26-defficient ribosomes in RAN translation. Therefore, we have toned down our claims. Now, we state that the RPS26/25/TSR2 insufficiency or depletion affects RAN translation. We also changed the title of the manuscript to: “Insufficiency of 40S ribosomal proteins, RPS26 and RPS25, negatively affects biosynthesis of polyglycine-containing proteins in fragile-X associated conditions” Previously it was: “Ribosomal composition affects the noncanonical translation and toxicity of polyglycine-containing proteins in fragile X-associated conditions”.

Specific points:(1) Analysis of the mass spec data in Supplemental Table S3 indicates that for many of the proteins that are differentially enriched in one sample, a single peptide is identified. So the difference is between 1 peptide and 0. I don't understand how one can do a statistical analysis on that, or how it would give out anything of significance. I certainly do not think it is significant. This is exacerbated by the fact that the contaminants in the assay (keratins) are many, many-fold more abundant, and so are proteins that are known to be mitochondrial or nuclear, and therefore likely not actual targets (e.g. MCCC1, PC, NPM1; this includes many proteins "of significance" in Table S1, including Rrp1B, NAF1, Top1, TCEPB, DHX16, etc...).The data in Table S6/Figure 3A suffer from the same problem.I am not convinced that the mass spec data is reliable.

We thank Reviewer for the comment concerning MS data; however, we believe that it may stem from misunderstanding of the data presented in Table S3 and S6. Both tables represent the output from MaxQuant analysis (so-called ProteinGroup) of MS .raw files, without any filtering. As stated in the Material&Methods, we applied default parameters suggested by MaxQuant developers to analyze MS data, these include identification of proteins based on at least 1 unique peptide, and thus some of the proteins with only 1 unique peptide are shown in Tables S1 and S3. Reviewer is also right that in this output table common contaminants, such as keratins are included. However, these identifications are denoted as “CON_”, and are further filtered out during statistical analysis in Perseus software. During the statistical analysis we first filtered out irrelevant protein groups identifications, such as contaminants, or only identified by site modifications.

We have changed the names of Supplementary Table files, giving more detailed description. We hope this will help to avoid misunderstanding for broader public. Secondly, when comparing the data presented in Table S3 and volcano plot presented in Figure 1B, one can notice that indeed the majority of identified proteins are not statistically significant (grey points), thus not selected for further stratification. Lack of significance of these proteins may be partially due to poor MS identification, however, they are not included in the following parts of the manuscript. Further, we selected only eight proteins (out of over 150) for stratification by orthogonal techniques, thus we argue that this step validates the biological relevance of chosen candidate RAN-translation modifiers. One should also keep in mind that pull down samples analyzed by MS often yield lower intensity and identification rates, when comparing to whole cell analysis, as a result of lower protein input or stringent washes used during sample preparation.

Regarding the data presented in Table S6 (SILAC data), we argue that these data are of very good quality. More than 2,000 proteins were identified in a 125min gradient, with over 80% of proteins that were identified with at least 2 unique peptides. Each of three biological replicates was analyzed three times (technical replicates), giving total of 9 high resolution MS runs. Together, we strongly believe that this data is of high confidence.

(2) The mass-spec data however claims to identify Rps26 as a factor binding the toxic RNA specifically. The rest of the paper seeks to develop a story of how Rps26-deficient ribosomes play a role in the translation of this RNA. I do not consider that this makes sense.

Indeed, we identified RPS26 as a protein that co-precipitated with *FMR1* containing expanded CGG repeats (Supplementary Figure 1G) and found that depletion of RPS26 hindered RAN translation of FMRpolyG, suggesting that RPS26 positively affects RAN translation. However, we did not state that RPS26 directly interacts with toxic RNA. In order to confirm the specificity of RAN translation regulation by RPS26 insufficiency, we tested whether depletion of other 40S ribosomal protein, RPS6, affects FMRpolyG synthesis. Our experiments showed that there was no any significant effect on RAN translation efficiency post RPS6 silencing (new Figure 5C). Importantly, we showed that RPS26 depletion did not inhibit global translation (new Figure 2B). In addition, mutagenesis of near-cognate start codon (new Figure 4F) and ASO_ACG treatment (new Figure 4G) provided the evidences that modulation of FMRpolyG biosynthesis by RPS26 level may depend on start codon selection. In essence, our data suggest that RPS26 depletion specifically affects synthesis of FMRpolyG, but not FMRP derived from the same *FMR1* mRNA with CGGexp. However, we do not claim that the observed effect is the consequence of a direct interaction between RPS26 and 5’UTR of FMR1 mRNA. Downregulation of FMRpolyG biosynthesis could be an outcome of the alteration of ribosomal assembly, decrease of efficiency and fidelity of PIC scanning/initiation or impeded elongation or a combination of all these processes. In the manuscript we presented the results of experiments which tested many of these possibilities.

(3) Rps26 is an essential gene, I am sure the same is true for DHX15. What happens to cell viability? Protein synthesis? The yeast experiments were carefully carried out under experiments where Rps26 was reduced, not fully depleted to give small growth defects.

We agree with the Reviewer that RPS26 and DHX15 are essential proteins, similarly to all RNA binding proteins, and caution should be taken during experimental design. To address this, we titrated different concentrations of siRPS26, and found that administration of 5 nM siRPS26, which just partially silenced RPS26, decreased FMRpolyG by around 50% (new Figure 1D). This impact was even greater with 15 nM siRPS26, as we observed around 80% decrease of FMRpolyG.

Havkin-Solomon et al. (2023), showed that proliferation rate is decreased in cells with mutated C-terminus of RPS26, which is required for contacting mRNA. In accordance with this study, we showed that cells with knocked down RPS26 proliferate less efficiently (new Figure 2A), but depletion of RPS26 did not impact the global translation (new Figure 2B). In addition, our SILAC-MS data indicates that ~80% of proteins with determined expression level were not affected by RPS26 insufficiency, and ~20% of the proteins turned out to be sensitive to RPS26 decrease. Although, these data do not take into account the protein stability.

(4) Knockdown efficiency for all tested genes must be shown to evaluate knockdown efficiency.

The current version of the manuscript contains representative western blots with validation of knock-down efficiency (for example in Figure 3B, C, E, Figure 6A) and we included knock-down validations where applicable (Figures 1D, 2B, 4G and 5C).

(5) The data in Figure 1E have just one mock control, but two cell types (control si and Rps26 depletion).

Mock control corresponds to the cells treated with lipofectamine reagent and was included in the study to determine the “background” signal from cells treated with delivery agent and reagents used to measure the apoptosis process. These cells were neither expressing FMRpolyG, nor siRNAs. Luminescence signals were normalized to the values obtained from mock control. We added more details describing this assay in the Figure 1 legend.

(6) The authors' data indicate that the effects are not specific to Rps26 but indeed also observed upon Rps25 knockdown. This suggests strongly that the effects are from reduced ribosome content or blocked protein synthesis. Additional controls should deplete a core RP to ascertain this conclusion.

We agree that observed effects may stem from reduced ribosome content, however, we argue that this is the only possibility and explanation. Previously, it was shown that RPS25 regulates G4C2-related RAN translation, but knock out of RPS25 does not affect global translation (Yamada S, 2019, *Nat. Neuroscience*). Similarly, we showed that KD of RPS26 or TSR2 did not reduce significantly global translation rate (SUnSET assay; new Figure 2B and 5C, respectively).

Moreover, in a new version of manuscript we included a control experiment, where we silenced core ribosomal protein (RPS6) and found that RPS6 depletion did not affect RAN translation from mutant FMR1 mRNA (new Figure 5C), thus strengthening our conclusion about specific RAN translation regulation by the level of RPS26 and RPS25.

Finally, our observation aligns well with current knowledge about how deficiency of different ribosomal proteins alters translation of some classes of mRNAs (Luan Y, 2022, *Nucleic Acids Res*; Cheng Z, 2019, *Mol Cell*). It was shown that depletion of RPS26 affects translation rate of different mRNAs compared to depletion of other proteins of small ribosomal subunit.

(7) Supplemental Figure S3 demonstrates that the depletion of S26 does not affect the selection of the start codon context. Any other claim must be deleted. All the 5'-UTR logos are essentially identical, indicating that "picking" happens by abundance (background).

Supplementary Figure 3D represents results indicating that the mutation in -4 position (from G to A) did not affect the RAN translation regardless of RPS26 presence or depletion. However, this result does not imply that RPS26 does not affect the selection of start codon of sequence- or RNA structure-context. We verified this particular -4 position, as it was suggested previously as important RPS26-sensitive site in yeasts (Ferretti M, 2017, *Nat Struct Mol Biol*). We agree with Reviewer that all 5’UTR logos presented in our paper did not show statistical significance for neither tested position for human mRNAs. On the contrary, we observed that regulation sensitive to RPS26 level depends on the selection of start codon of RAN translation, in particular ACG initiation (new Figure 4F&G). RPS26 depletion affected ACG-initiated but not GTG- or CTG-initiated RAN translation.

In the previous version of the manuscript, we wrote that we did not identify any specific motifs or enrichment within analyzed transcripts in comparison to the background. On the other hand, we found that the GC-content among analyzed transcripts is higher within 5’UTRs and in close proximity to ATG in coding sequences (Figure 4D), what suggests the importance of RNA stable structures in this region. In addition, we showed that mRNAs encoding proteins responding to RPS26 depletion have shorter than average 5’UTRs (new Figure 4E).

(8) Mechanism is lacking entirely. There are many ways in which ribosomes could have mRNA-specific effects. The authors tried to find an effect from the Kozak sequence, unsuccessfully (however, they also did not do the experiment correctly, as they failed to recognize that the Kozak sequence differs between yeast, where it is A-rich, and mammalian cells, where it is GGCGCC). Collisions could be another mechanism.

Indeed, collisions as well as other mechanisms such as skewed start codon fidelity may have an effect on efficiency of FMRpolyG biosynthesis. In the current version of the manuscript, we show that RPS26 amount-sensitive regulation seems to be start codonselection dependent (new Figure 4F&G).

**Reviewer #2 (Public Review):**
Summary:Translation of CGG repeats leads to the accumulation of poly G, which is associated with neurological disorders. This is a valuable paper in which the authors sought out proteins that modulate RAN translation. They determined which proteins in Hela cells bound to CGG repeats and affected levels of polyG encoded in the 5'UTR of the FMR1 mRNA. They then showed that siRNA depletion of ribosomal protein RPS26 results in less production of FMR1polyG than in control. There are data supporting the claim that RPS26 depletion modulates RAN translation in this RNA, although for some results, the Western results are not strong. The data to support increased aggregation by polyG expression upon S26 KD are incomplete.

We thank the Reviewer for critical comments and suggestions. We sincerely appreciate your rigorous, insightful, and constructive feedback throughout the revision process.

Below each specific point, we addressed the mentioned issues.

Strengths:The authors have proteomics data that show the enrichment of a set of proteins on FMR1 RNA but not a related RNA.

We thank Reviewer for appreciation of provided MS-screening results, which identified proteins enriched on FMR1 RNA with expanded CGG repeats.

Weaknesses:- It is insinuated that RPS26 binds the RNA to enhance CGG-containing protein expression. However, RPS26 reduction was also shown previously to affect ribosome levels, and reduced ribosome levels can result in ribosomes translating very different RNA pools.

In previous version of the manuscript we did not state that RPS26 binds directly to RNA with expanded CGG repeats and we did not show the experiment indicating direct interaction between studied RNA and RPS26. What we showed is that RPS26 was enriched on *FMR1* RNA MS samples, however, we did not verify whether it is direct or indirect interaction. We also tried to test hypothesis that lack of RPS26 in PIC complex may affect efficiency of RAN translation initiation via specific, previously described in yeast Kozak context (Ferretti M, 2017, *Nat Struct Mol Biol*). As we described this hypothesis was negatively validated. However, we showed that other features of 5’UTR sequences (e.g. higher GC-content or shorter leader sequence) are potentially important for translation efficiency in cells with depleted RPS26.

Indeed, RPS26 is involved in 40S maturation steps (Plassart L, 2021, *eLife*) and its insufficiency or mutations or blocking its inclusion to 40S ribosome may result in incomplete 40S maturation, which subsequently might negatively affect translation per se. However, we did not observe global translation inhibition after RPS26 depletion or depletion of TSR2, the chaperon involved in incorporation/exchange RPS26 to small ribosomal subunit (new Figure 2B and 5C). In addition, our SILAC-MS data indicates that majority of studied proteins (including FMRP, the main product of *FMR1* gene) were not affected by RPS26 depletion which can be carefully extrapolated to global translation. In revised manuscript we also showed that relatively low silencing of RPS26 also decreased FMRpolyG production in model cells (new Figure 1D).

We agree that reduced ribosome levels can result in different efficiency of translation of different RNA pools. We enhance this statement in revised manuscript. However, we also showed that the same mRNA containing different near cognate start codons (single/two nucleotide substitution) specific to RAN translation, or targeting this codon with antisense oligonucleotides resulted in altered sensitivity of FMR1 mRNA translation to RPS26 depletion (new Figure 4F).

- A significant claim is that RPS26 KD alleviates the effects of FMRpolyG expression, but those data aren't presented well.

We thank the Reviewer for this comment. In the new version of the manuscript, we have added new microscopic images and improved the explanation of Figure 1E. We have also completed the interpretation of Figure 1F in the main text, figure image as well as figure legend, and we hope that these changes will ameliorate understanding of our data.

**Recommendations For The Authors:**
- A significant claim is that RPS26 KD alleviates the effects of FMR polyG expression, but those data aren't presented well:Figure 1D (supporting data in S2) and 2D - the authors need to show representative images of a control that has aggregation and indicate aggregates being counted on an image. The legend states that there are no aggregates, but the quantification of aggregates/nucleus is ~1, suggesting there are at least 1 per cell. It is preferred to show at least a representative of what is quantified in the main figure instead of a bar graph.

The representative images of control and siRPS26-treated cells are now shown in revised version of Figure 1E. Additionally, we completed the Figure legend concerning this part, as well as extended description of the experiment in Materials&Methods section.

Figure 1E - it is unclear what luminescence signal is being measured. Is this a dye for an apoptotic marker? More information is needed in the legend.

This information was added to the legend of modified Figure 1F (previously 1E) as suggested.

- Some of the Western blots are not very convincing. Better evidence for the changes in bar graphs would improve how convincing the data are:Fig 2B. The western for FMR95G in the first model is not very convincing. The difference by eye for the second siRNA seems to give a larger effect than the first for 95G construct but they appear almost the same on the graph. More supporting information for the quantification is needed.

We provided better explanation for WB quantification in M&M section in the manuscript. Alos, we provided additional blot demonstrating independent biological replicate of the mentioned experiment in supplementary materials (Supplementary Figure S2E).

Figure 4A, the blots for RPS26 and FMR95G are not convincing. They are quite smeary compared to all of the others shown for these proteins in other figures. Could a different replicate be shown?

We provided additional blot demonstrating the effect on transiently expressed FMRpolyG affected by depletion of TSR2 in COS7 cell line (Supplementary Figure S4A).

Figure 5A and 5B blots are not ideal. Could a different replicate be shown? Or show multiple replicates in the supplemental figure?

We provided additional blots from the same experiment, although data is not statistically significant, most likely due to low quality of normalization factor, which is Vinculin (Supplementary Figure S5A). Nevertheless, the level of FMRpolyG is decreased by ~70% after RPS25 silencing in SH-SY5Y cells.

Figure 2C. Please use the same y axes for all four Westerns in B and C. One would like to compare 95 and 15 repeats, but it is difficult when the y axes are different.

Thank you for this comment. The y axis was adjusted as suggested by the Reviewer.

Figure 3D-The text suggests a significant difference between positive and negative responders that is not clear in the figure.

In the main body of the manuscript we state that: “We did not observe any significant differences in the frequency of individual nucleotide positions in the 20-nucleotide vicinity of the start codon relative to the expected distribution in the BG”, which is in line with the graph showed in Figure 4D (previously 3D).

**Reviewer #3 (Public Review):**
Tutak et al provide interesting data showing that RPS26 and relevant proteins such as TSR2 and RPS25 affect RAN translation from CGG repeat RNA in fragile X-associated conditions. They identified RPS26 as a potential regulator of RAN translation by RNAtagging system and mass spectrometry-based screening for proteins binding to CGG repeat RNA and confirmed its regulatory effects on RAN translation by siRNA-based knockdown experiments in multiple cellular disease models and patient-derived fibroblasts. Quantitative mass spectrometry analysis found that the expressions of some ribosomal proteins are sensitive to RPS26 depletion while approximately 80% of proteins including FMRP were not influenced. Since the roles of ribosomal proteins in RAN translation regulation have not been fully examined, this study provides novel insights into this research field. However, some data presented in this manuscript are limited and preliminary, and their conclusions are not fully supported.(1) While the authors emphasized the importance of ribosomal composition for RAN translation regulation in the title and the article body, the association between RAN translation and ribosomal composition is apparently not evaluated in this work. They found that specific ribosomal proteins (RPS26 and RPS25) can have regulatory effects on RAN translation (Figures 1C, 2B, 2C, 2E, 4A, 5A, and 5B), and that the expression levels of some ribosomal proteins can be changed by RPS26 knockdown (Figure 3B, however, the change of the ribosome compositions involved in the actual translation has not been elucidated). Therefore, their conclusive statement, that is, "ribosome composition affects RAN translation" is not fully supported by the presented data and is misleading.

We thank the Reviewer for critical comments and suggestions. We agree that the initial title and some statements in the text were misleading and the presented data did not fully support the aforementioned statement regarding ribosomal composition affecting FMRpolyG synthesis. Therefore, in the revised version of the manuscript we included a control experiment indicating that depletion of another core 40S ribosomal protein (RPS6) did not impact the FMRpolyG synthesis (new Figure 5C), which supports our hypothesis that RPS26 and RPS25 are specific CGG-related RAN translation modifiers. To precisely deliver a main message of our work, we changed the title that will indicate the specific effect of RPS26 and RPS25 insufficiency on RAN translation of FMRpolyG. Proposed title: “Insufficiency of 40S ribosomal proteins, RPS26 and RPS25 negatively affects biosynthesis of polyglycine-containing proteins in fragile-X associated conditions”. We also changed all statements regarding “ribosomal composition” in main text of the new version of manuscript.

(2) The study provides insufficient data on the mechanisms of how RPS26 regulates RAN translation. Although authors speculate that RPS26 may affect initiation codon fidelity and regulate RAN translation in a CGG repeat sequence-independent manner (Page 9 and Page 11), what they really have shown is just identification of this protein by the screening for proteins binding to CGG repeat RNA (Figure 1A, 1B), and effects of this protein on CGG repeat-RAN translation. It is essential to clarify whether the regulatory effect of RPS26 on RAN translation is dependent on CGG repeat sequence or near-cognate initiation codons like ACG and GUG in the 5' upstream sequence of the repeat. It would be better to validate the effects of RPS26 on translation from control constructs, such as one composed of the 5' upstream sequence of FMR1 with no CGG repeat, and one with an ATG substitution in the 5' upstream sequence of FMR1 instead of near-cognate initiation codons.

We agree that the data presented in the manuscript implies that insufficiency of RPS26 plays a pivotal role in the regulation of CGG-related RAN translation and in the revised version of the manuscript we included a series of experiments indicating that ACG codon selection seems to be an important part of RPS26 level-dependent regulation of polyglycine production (new Figure 4F&G; see point 3 below for more details). Importantly, in the luciferase assay showed on Figure 4F we used the AUG-initiated firefly luciferase reporter as normalization control.

Moreover, to verify if FMRpolyG response to RPS26 deficiency depends on the type of reporter used, we repeated many experiments using FMRpolyG fused with different tags. The luciferase-based assays were in line with experiments conducted on constructs with GFP tag (new Figure 1D), thus strengthening our previous data. Moreover, in the series of experiments, we show that FMRP synthesis which is initiated from ATG codon located in *FMR1* exon 1, was not affected by RPS26 depletion (Figure 3E & 4C), even though its translation occurs on the same mRNA as FMRpolyG. This indicates a specific RPS26 regulation of polyglycine frame initiated from ACG near cognate codon.

(3) The regulatory effects of RPS26 and other molecules on RAN translation have all been investigated as effects on the expression levels of FMRpolyG-GFP proteins in cellular models expressing CGG repeat sequences Figures 1C, 2B, 2C, 2E, 4A, 5A, and 5B. In these cellular experiments, there are multiple confounding factors affecting the expression levels of FMRpolyG-GFP proteins other than RAN translation, including template RNA expression, template RNA distribution, and FMRpolyG-GFP protein degradation. Although authors evaluated the effect on the expression levels of template CGG repeat RNA, it would be better to confirm the effect of these regulators on RAN translation by other experiments such as in vitro translation assay that can directly evaluate RAN translation.

We agree that there are multiple factors affecting final levels of FMRpolyG-GFP proteins including aforementioned processes. We evaluated the level of *FMR1* mRNA, which turned out not to be decreased upon RPS26 depletion (Figure 3B&C), therefore, we assumed that what we observed, was the regulation on translation level, especially that RPS26 is a ribosomal protein contacting mRNA in E-site. We believe that direct assays such as in vitro translation may be beneficial, however, depletion of RPS26 from cellular lysate provided by the vendor seems technically challenging, if not completely impossible. Instead, we focused on sequence/structure specific regulation of RAN translation with the emphasis on start-codon initiation selection. It resulted in generating the valuable results pointing out the RPS26 role in start codon fidelity (Figure 4F&G). These new results showed that translation from mRNAs differing just in single or two nucleotide substitution in near cognate start codon (ACG to GUG or ACG to CUG), although results in exactly the same protein, is differently sensitive to RPS26 silencing (new Figure 4F). Similar differences were observed for translation efficiency from the same mRNA targeted or not with antisense oligonucleotide complementary to the region of RAN translation initiation codon (new Figure 4G). These results also suggest that stability of FMRpolyG is not affected in cells with decreased level of RPS26.

(4) While the authors state that RPS26 modulated the FMRpolyG-mediated toxicity, they presented limited data on apoptotic markers, not cellular viability (Figure 1E), not fully supporting this conclusion. Since previous work showed that FMRpolyG protein reduces cellular viability (Hoem G, 2019,Front Genet), additional evaluations for cellular viability would strengthen this conclusion.

We thank the Reviewer for this suggestion. We addressed the apoptotic process in order to determine the effect of RPS26 depletion on RAN translation related toxicity (Figure 1F). In revised version of the manuscript, we also added the evaluation on how cells proliferation was affected by RPS26, RPS25, RPS6 and TSR2 depletion. Our data indicate that TSR2 silencing slightly impacted the cellular fitness (new Figure 5D), whereas insufficiencies of RPS26, RPS25 and RPS6 had a much stronger negative effect on proliferation (new Figure 2A, 5D, 6C), which is in line with previous data (Cheng Z 2019, *Mol Cell*; Luan Y, 2022, *Nucleic Acids Res)*. The difference in proliferation rate after treatment with siRPS26 makes proper interpretation of cellular viability assessment very difficult.

**Recommendations For The Authors:**
(1) It would be nice to validate the effects of overexpression of RPS26 and other regulators on RAN translation, not limited to knockdown experiments, to support the conclusion.

We did not performed such experiments because we believed that RPS26 overexpression may have no or marginal effect on translation or RAN translation. It is likely impossible to efficiently incorporate overexpressed RPS26 into 40S subunits, because the concentration of all ribosomal proteins in the cells is very high.

(2) It would be better to explain how authors selected 8 proteins for siRNA-based validation (Figure 1C, 1D, S1D) from 32 proteins enriched in CGG repeat RNA in the first screening.

We selected those candidates based on their functions connected to translation, structured RNA unwinding or mRNA processing. For example, we tested few RNA helicases because of their known function in RAN translation regulation described by other researchers. This explanation was added to the revised version of the manuscript.

(3) Original image data showing nuclear FMRpolyG-GFP aggregates should be presented in Figure 1D.

The representative images of control and siRPS26-treated cells are now shown in modified version of Figure 1E and described with more details in the legend.

(4) Image data in Figure 2A and 2D have poor signal/noise ratio and the resolution should be improved. In addition, aggregates should be clearly indicated in Figure 2D in an appropriate manner.

The stable S-FMR95xG cellular model is characterized by very low expression of RANtranslated FMR95xG, therefore, it is challenging to obtain microscopic images of better quality with higher GFP signal. In the L-99xCGG model expression of transgene is higher. Therefore, we provided new image in the new version of Figure 3D (former 2D). Moreover, we showed aggregates on the image obtained using confocal microscopy (new Supplementary Figure 2D).

(5) The detailed information on patient-derived fibroblast (age and sex of the patient, the number of CGG repeats, etc.) in Figure 2F needed to be presented.

This information was added to the figure legend (Figure 3F; previously 2F) and in the Material and Methods section as suggested.

(6) It would be better to normalize RNA expression levels of FMR1 and FMR1-GFP by the housekeeping gene in Figure S2C, like other RT-qPCR experimental data such as Figure 2B.

Normalization of FMR1-GFP to GAPDH is now shown in modified version of Figure S2C (right graph) as requested by the Reviewer.

(7) It would be better to add information on molecular weight on all Western blotting data.(8) Marks corresponding to molecular weight ladder were added to all images.

Full blots, including protein ladders were deposited in Zenodo repository, under doi: 10.5281/zenodo.13860370

References

Cheng Z, Mugler CF, Keskin A, Hodapp S, Chan LYL, Weis K, Mertins P, Regev A, Jovanovic M & Brar GA (2019) Small and Large Ribosomal Subunit Deficiencies Lead to Distinct Gene Expression Signatures that Reflect Cellular Growth Rate. *Mol Cell* 73: 36-47.e10

Havkin-Solomon T, Fraticelli D, Bahat A, Hayat D, Reuven N, Shaul Y & Dikstein R (2023) Translation regulation of specific mRNAs by RPS26 C-terminal RNA-binding tail integrates energy metabolism and AMPK-mTOR signaling. *Nucleic Acids Res* 51: 4415–4428

Hoem,G., Larsen,K.B., Øvervatn,A., Brech,A., Lamark,T., Sjøttem,E. and Johansen,T. (2019) The FMRpolyGlycine protein mediates aggregate formation and toxicity independent of the CGG mRNA hairpin in a cellular model for FXTAS. Front. Genet., 10, 1–18.

Luan Y, Tang N, Yang J, Liu S, Cheng C, Wang Y, Chen C, Guo YN, Wang H, Zhao W, *et al* (2022) Deficiency of ribosomal proteins reshapes the transcriptional and translational landscape in human cells. *Nucleic Acids Res* 50: 6601–6617

Plassart L, Shayan R, Montellese C, Rinaldi D, Larburu N, Pichereaux C, Froment C, Lebaron S, O’donohue MF, Kutay U, *et al* (2021) The final step of 40s ribosomal subunit maturation is controlled by a dual key lock. *Elife* 10